# Ouabain Ameliorates Alzheimer’s Disease-Associated Neuropathology and Cognitive Impairment in FAD^4T^ Mice

**DOI:** 10.3390/nu16203558

**Published:** 2024-10-20

**Authors:** Dan Wang, Jiajia Liu, Qizhi Zhu, Xin Wei, Xiang Zhang, Qi Chen, Yu Zhao, Heng Tang, Weiping Xu

**Affiliations:** 1Division of Life Sciences and Medicine, University of Science and Technology of China, Hefei 230026, China; danw2020@mail.ustc.edu.cn (D.W.); liujiajia@mail.ustc.edu.cn (J.L.); xz0826@mail.ustc.edu.cn (X.Z.); chenqi421@mail.ustc.edu.cn (Q.C.); hfzhaoyu@163.com (Y.Z.); tangheng@mail.ustc.edu.cn (H.T.); 2Anhui Provincial Key Laboratory of Tumor Immunotherapy and Nutrition Therapy, Hefei 230001, China; 3The Science Island Branch of Graduate School, University of Science and Technology of China, Hefei 230026, China; zqz2020@mail.ustc.edu.cn (Q.Z.); weixin8299@mail.ustc.edu.cn (X.W.)

**Keywords:** ouabain, Alzheimer’s disease, cognitive impairment, microglia, TREM2

## Abstract

**Background:** Alzheimer’s disease (AD) is a common clinical neurodegenerative disorder, primarily characterized by progressive cognitive decline and behavioral abnormalities. The hallmark pathological changes of AD include widespread neuronal degeneration, plaques formed by the deposition of amyloid β-protein (Aβ), and neurofibrillary tangles (NFTs). With the acceleration of global aging, the incidence of AD is rising year by year, making it a major global public health concern. Due to the complex pathology of AD, finding effective interventions has become a key focus of research. Ouabain (OUA), a cardiac glycoside, is well-known for its efficacy in treating heart disease. Recent studies have also indicated its potential in AD therapy, although its exact mechanism of action remains unclear. **Methods:** This study integrates bioinformatics, multi-omics technologies, and in vivo and in vitro experiments to investigate the effects of OUA on the pathophysiological changes of AD and its underlying molecular mechanisms. **Results:** This study analyzed the expression of the triggering receptor expressed on myeloid cells 2 (TREM2) across different stages of AD using bioinformatics. Serum samples from patients were used to validate soluble TREM2 (sTREM2) levels. Using an Aβ_1-42_-induced microglial cell model, we confirmed that OUA enhances the PI3K/AKT signaling pathway activation by upregulating TREM2, which reduces neuroinflammation and promotes the transition of microglia from an M1 proinflammatory state to an M2 anti-inflammatory state. To evaluate the in vivo effects of OUA, we assessed the learning and memory capacity of FAD^4T^ transgenic mice using the Morris water maze and contextual fear conditioning tests. We used real-time quantitative PCR, immunohistochemistry, and Western blotting to measure the expression of inflammation-associated cytokines and to assess microglia polarization. OUA enhances cognitive function in FAD^4T^ mice and has been confirmed to modulate microglial M1/M2 phenotypes both in vitro and in vivo. Furthermore, through bioinformatics analysis, molecular docking, and experimental validation, TREM2 was identified as a potential target for OUA. It regulates PI3K/Akt signaling pathway activation, playing a crucial role in OUA-mediated M2 microglial polarization and its anti-inflammatory effects in models involving Aβ_1-42_-stimulated BV-2 cells and FAD^4T^ mice. **Conclusions:** These findings indicate that OUA exerts anti-neuroinflammatory effects by regulating microglial polarization, reducing the production of inflammatory mediators, and activating the PI3K/Akt signaling pathway. Given its natural origin and dual effects on microglial polarization and neuroinflammation, OUA emerges as a promising therapeutic candidate for neuroinflammatory diseases such as AD.

## 1. Introduction

Alzheimer’s disease (AD), the most common form of dementia, is characterized by progressive memory loss and cognitive deficits. Its main pathological mechanisms include the degeneration and aggregation of β-amyloid (Aβ) and the development of neurofibrillary tangles (NFTs) formed by Tau protein [1,2]. It is estimated that by 2050, the number of dementia patients worldwide will increase to 153 million. As the global population ages rapidly, AD has emerged as a critical public health crisis, driving the urgent need for effective treatments.

The pathogenesis of AD is intricate, and current drug therapies offer only limited efficacy. Growing evidence highlights the pivotal role of inflammation in AD pathogenesis, suggesting that neuroinflammation is a significant disease mechanism [3,4]. Inflammation is a key feature of many diseases, including neurodegenerative diseases, making the inhibition of inflammation a common therapeutic strategy [5]. Within the central nervous system, microglia, which act as dynamic surveillance cells, play crucial roles in tissue maintenance, the injury response, and pathogen defense [5,6,7]. Microglia exhibit significant functional plasticity in central nervous system (CNS) diseases [8]. When stimulated, microglia can be activated into M1 proinflammatory or M2 anti-inflammatory phenotypes, which are responsible for clearing microbes, dead cells, excess synapses, protein aggregates, and other particles and soluble antigens that could pose a threat to the central nervous system [9,10,11]. Promoting the transition of M1 microglia to M2 microglia, thereby exerting neuroprotective effects and suppressing central inflammatory responses, is a crucial goal. Most of the genes associated with AD identified in genome-wide association studies are highly expressed in or exclusive to microglia [12]. The triggering receptor expressed on the myeloid cells 2 (*TREM2*) gene is situated on chromosome 6p21.1 and within an immune function gene cluster on mouse chromosome 17C3 [13]. In the periphery, it is mainly found at the myeloid cells surface, including dendritic cells, granulocytes, and macrophages. Conversely, within the central nervous system, TREM2 is solely present on microglial surfaces [14]. Rare mutations in the *TREM2* gene elevate the risk of late-onset AD by approximately threefold, making it one of the most highly known risk genes for AD [15]. Therefore, *TREM2* is considered an important biomarker and potential therapeutic target for AD.

Ouabain (OUA), also known as G-strophanthin, is a cardiac glycoside drug. It was initially extracted from the roots, stems, leaves, and seeds of East African plants such as *Acokanthera schimperi* [16,17] and *Strophanthus gratus* [18,19]. In various Kenyan tribes, ouabain was widely used as an arrow poison for hunting and warfare [20]. The Food and Drug Administration (FDA) approved OUA for clinical treatment of heart failure and arrhythmias. Its mechanism involves inhibiting Na+/K+-ATPase activity to produce a positive inotropic effect [21,22,23]. The earliest report on the anti-inflammatory properties of ouabain came from Lancaster and Vegad [24], who noted that ouabain could inhibit vascular permeability and reduce inflammation caused by pleural and skin irritations in sheep. Leite et al. [25] found that in a mouse model of yeast polysaccharide-induced peritonitis, ouabain lessened the secretion of inflammatory cytokines such as IL-1β and TNF-α, consequently reducing the severity of the acute condition. In various studies, OUA has shown neuroprotective effects. In an optic nerve transection model, ouabain activated the autophagy pathway, reduced the expression of receptors such as TNFR1/2, TLR4, and CD14 in the retinal cells of neonatal rats, inhibited neuroinflammation, and increased the neuroprotection of RGCs [26]. Sibarov et al. [27] reported that ouabain prevents excitotoxic neuronal cell death by modulating the interaction between the sodium–calcium exchanger (NCX) and N-methyl-D-aspartate receptors (NMDARs), as well as synaptic transmission. Yun Xu et al. [28] identified potential drugs for Alzheimer’s disease by utilizing risk genes from GWAS and single-cell RNA sequencing studies, discovering that ouabain showed notable anti-inflammatory and neuroprotective properties.

OUA’s dual anti-inflammatory and neuroprotective actions highlight its multifaceted potential in combating AD. Therefore, this study aimed to utilize network pharmacology, bioinformatics, and both in vitro and in vivo experimental validation to elucidate the role and mechanisms through which OUA modulates cognitive dysfunction in AD.

## 2. Materials and Methods

### 2.1. Chemicals and Reagents

Ouabain (>98%) (cat. no.11018-89-6) was purchased from Sigma (St. Louis, MO, USA). TREM2 (RRID: AB_11226344), GFAP (RRID: AB_305808), SYN (RRID: AB_2286526) and PSD95 RRID: AB_2864417) antibodies were obtained from Abcam (Cambridge, UK). Arginase-1 (RRID: AB_2109205), iNOS (RRID: AB_2725360), PI3K (RRID: AB_10694519), AKT RRIDAB: 2225340), p-PI3K (cat. no. CY36427), and p-AKT (cat. no. CY6016) were obtained from Proteintech (Wuhan, China). TRIzol (cat. no. 15596026), the BCA protein assay kit (cat. no. PC0020), and DMSO (cat. no. D8371) were obtained from Solarbio (Beijing, China). DMEM (cat. no. C0891), Cell Counting Kit-8 (cat. no. C0041), amyloid β peptide (_1-42_) (cat. no. P9001), the Calcein/PI cell Viability kit (cat. no. C2015S), and 0.05% Trypsin-EDTA solution (cat. no. C0202) were purchased from Beyotime (Shanghai, China). TREM2 ELISA kits (cat. no. RX101183H, Quanzhou, China) was obtained from Ruixin Bio; and fetal bovine serum (cat. no. A5669701, New York, NY, USA) was obtained from Gibco. We used BV2 cells (mouse microglial cells) purchased from Shanghai Fuheng Biotechnology Co., Ltd. for the cell experiments in this study.

### 2.2. Bioinformatics Analysis

Gene expression data from AD patients were obtained from Gene Expression Omnibus (GEO) datasets (GSE28146 and GSE5281). The GEO dataset GEO28146 contains sequencing data from brain tissue samples of 30 subjects. Based on MMSE and NFT values, the 30 subjects were divided into four groups: control group (8 subjects), early stage (7 subjects), moderate stage (8 subjects), and severe stage (7 subjects). For further analysis, the four groups were consolidated into two groups: severe group (7 subjects) and non-severe group (23 subjects). The GSE5281 dataset contains sequencing data from a total of 161 samples. Based on disease status, the 161 samples were divided into the normal group (74 samples) and Alzheimer’s disease group (87 samples). For further analysis, we examined the expression levels of TREM2 in the 87 Alzheimer’s disease samples. According to the median TREM2 expression value among these 87 AD patients, we divided them into two groups: high (43 subjects) and low (44 subjects). The R package “limma” was used to identify differentially expressed genes. For Gene Ontology (GO) and Kyoto Encyclopedia of Genes and Genomes (KEGG) (www.kegg.jp/kegg/kegg1.html, accessed on 12 June 2023) pathway enrichment analysis of all candidate differential genes, the R packages “clusterProfiler” and “org.Hs.eg.db” were employed [29]. A *p*-value threshold of <0.05 was set to determine significant pathway enrichment, and the findings were visualized with the R version 4.0.3 package “ggplot2”. Additionally, gene set enrichment analysis (GSEA) was performed to investigate the underlying pathway variations between the two groups. In our bioinformatics analysis, we applied multiple hypothesis testing correction to control the false positive rate, ensuring the robustness of the statistical inferences. Additionally, we adopted more stringent statistical significance thresholds and incorporated cross-validation methods to verify the reliability of the results, reducing the risk of overfitting. To further enhance the biological interpretability of the results, we also referred to biological background knowledge to filter the preliminary findings and eliminate potential misinterpretations caused by irrelevant discoveries.

### 2.3. Assay of sTREM2 Levels in Clinical Patients

Between March 2022 and August 2023, serum samples were collected from inpatients at the First Affiliated Hospital of the University of Science and Technology of China. The STREM2 detection was performed using the RuixinBio ELISA kit (Quzhou, China), with a detection range of 125–4000 pg/mL and a sensitivity of <10 pg/mL. According to the criteria set by the National Institute of Neurological and Communicative Disorders and Stroke–Alzheimer’s Disease and Related Disorders Association (NINCDS-ADRDA), these patients were diagnosed with “probable Alzheimer’s disease with mild to moderate dementia”. Fasting venous blood samples were obtained from all participants, and serum sTREM2 levels were assessed via ELISA. This study was approved by the hospital’s medical ethics committee (ethical approval number: 2023-ky030).

### 2.4. Predicting Active Ingredients and Targets of OUA 

The protein expression data were normalized using the UniProt protein database. Potential targets of OUA were predicted via Swiss Target Prediction (http://www.swisstargetprediction.ch/, accessed on 18 June 2023), while AD-related targets were sourced from the GeneCards (https://www.genecards.org/), OMIM database (https://www.omim.org/, accessed on 18 June 2023), and DrugBank database (https://www.drugbank.ca/, accessed on 18 June 2023).

### 2.5. GO and KEGG Enrichment Analyses

The common targets between OUA and AD were uploaded to the Metascape platform for GO and KEGG enrichment analyses. These analyses identified relevant biological processes and pathways, and the results were visualized using the online bioinformatics tool Microbioinfo.

### 2.6. Network Construction and Analysis

Data on potential OUA and AD targets were obtained and imported into Cytoscape 3.7.1 to construct detailed “drug–target–pathway” networks. The network employed distinct graphical nodes and colors to represent compounds, targets, and pathways, effectively providing a clear and intuitive visualization.

### 2.7. Molecular Docking of OUA with the TREM2 Protein

The OUA structure is obtained in SDF format from the PubChem database and then imported into ChemDraw3D. The MM2 module was used for energy minimization and hydrogenation, and charges were added using AutoDockTools 1.5.6. We obtained the 3D structure of the TREM2 protein from the PDB database. Using PyMOL, the original ligand water molecules were removed, followed by the addition of hydrogen atoms and the calculation of Gasteiger charges with AutoDockTools 1.5.6. The ligand was then docked with the receptor using AutoDock Vina 1.1.2, and the binding mode was analyzed and visualized using PyMOL 2.4.

### 2.8. Animal Groups and Intervention Methods

Six male C57 mice (SPF level) and twelve 6-month-old male FAD^4T^ mice (SPF level) were purchased from Jicui Yaokang Company (Nanjing China) and housed at the Laboratory Animal Center of USTC. In previous studies, the number of mice per group typically ranged from 3 to 5. Based on these precedents, we included 6 mice per group in this study and performed a power analysis using G*Power 3.0 software. We set the significance level at 0.05, the effect size at 0.8, and the statistical power at 0.78. The animal experiments were approved by the Animal Ethics Committee of the First Affiliated Hospital of the University of Science and Technology of China (ethical approval number: 2020-N(A)-298). All animal experimental procedures strictly adhered to the guidelines of the Laboratory Animal Management Regulations of the University. The mice were placed at a temperature of 18–22 °C and humidity of 50–60%. All mice were housed under SPF conditions with weekly changes of fresh food and water and maintained on a light/dark cycle (12 h/12 h). Fresh bedding was regularly added to the ventilated cage and small items were regularly sterilized. The mice were assigned to three groups of six each: a normal group, a model group, and a drug intervention group. FAD^4T^ mice in the drug intervention group received intraperitoneal injections of 2 µg/kg OUA at a slightly modified dose from the previous study. We used DMSO to ensure the dissolution of OUA powder and diluted it to the appropriate concentration with sterile saline for administration, while the normal and model groups were injected with an equal amount of saline three times a week for four weeks. After completing all behavioral experiments, the mice were euthanized using carbon dioxide (CO_2_) asphyxiation. To further ensure the effectiveness of euthanasia and comply with ethical standards for animal experiments, cervical dislocation was performed following CO_2_ exposure to confirm the death of the mice.

### 2.9. Morris Water Maze (MWM)

In this experiment, we used the Morris Water Maze (MWM) to assess the spatial learning and memory abilities of mice. Titanium dioxide was added to the water before the experiment, turning the water milky white to ensure that the mice could not see the hidden platform underwater. The water depth was set to 30 cm, and the temperature was maintained at 22 ± 2 °C. We used the Morris Water Maze analysis system from Beijing Zhongshi, with an overhead camera to record the swimming paths and the time it took the mice to find the platform. The experiment lasted 6 days and included three phases: First, in the pre-training phase, the mice were placed in different positions in the pool to familiarize themselves with the environment, with the experimenter guiding them to the platform. Next, in the training phase, the mice were placed in the water from quadrants A1, A2, and A4 to find the hidden platform in quadrant A3, and the escape latency (time to find the platform) was recorded. Finally, in the testing phase, the platform was removed, and the mice were placed in the water from quadrant A1. Their behavior in quadrant A3, including the number of crossings and time spent, was recorded to evaluate their spatial learning and memory abilities.

### 2.10. Fear Conditioning Experiment

This experiment utilized the Labmaze conditioned fear experiment system from Beijing Zhongshi, which can automatically control the output of a multifunctional stimulator, using sound as a cue. The system identifies three behavioral states of the mice, mobility, immobility, and freezing, and records and analyzes their behavior and movement trajectories. The experiment included four phases—adaptation, training, contextual fear memory testing, and cued memory testing. In the adaptation phase, mice were placed in a fear conditioning box for 5 min to acclimate to the new environment. On the second day, during the training phase, the mice acclimated in the box for 2 min before being exposed to a 30 s auditory cue (85 dB, 3000 Hz). A 0.75 mA foot shock was delivered at the final 2 s of the sound. This sequence was repeated 3 to 5 times with 70 s intervals, totaling 5 min. On the third day, during the contextual fear memory testing phase, the mice were kept in the box for 5 min without exposure to any cues or shocks, and freezing time was recorded for 5 min. On the fourth day, during the cued fear memory testing phase, after a 2 min adaptation period, a 30 s auditory cue was presented without a foot shock, and freezing time was recorded for 5 min.

### 2.11. Nissl Staining

After deparaffinization and hydration, we immersed the tissue sections in the Nissl staining solution for 10 min. To remove excess dye, we quickly differentiated the stained sections in 95% ethanol containing 0.5% acetic acid for a few seconds. After differentiation, we thoroughly rinsed the sections in distilled water until the background was clear. We then dehydrated the sections by transferring them through a graded series of ethanol solutions, and cleared the dehydrated sections by immersing them in xylene for 5 min, repeating this step twice. Finally, we mounted the sections in neutral resin to ensure long-term preservation.

### 2.12. Hematoxylin–Eosin (HE) Staining

Brain sections (6 μm thick) were deparaffinized by xylene and an ethanol gradient. The sections were then stained with hematoxylin, washed in distilled water, and differentiated using 1% hydrochloric acid in ethanol. After staining with eosin, the paraffin sections were dehydrated with anhydrous ethanol, cleared with xylene, and then observed under a light microscope.

### 2.13. Cell Culture and Processing

BV-2 cells (mouse microglia) were maintained in DMEM containing 10% fetal bovine serum, 100 U/mL of streptomycin, and 100 U/mL penicillin, and incubated at 37 °C with 5% CO_2_. When the cells reached approximately 90% confluence, they were digested with trypsin and then passaged.

### 2.14. Cell Viability Assay

BV-2 cells in the logarithmic growth phase were seeded into 96-well plates at 5 × 10^4^ cells/mL and incubated overnight to allow for adherence. Subsequently, various concentrations of OUA and Aβ_1-42_ were added to the 96-well plate, and the co-culture was maintained for 24 h. Next, 10 μL of the prepared CCK-8 working solution was added to each well and incubated at 37 °C in a 5% CO_2_ incubator for 2 h. After incubation, the 96-well plates were subsequently inserted into a microplate reader to assess the absorbance (OD value) at 450 nm for each well, which was used to calculate cell viability. Each group had at least three replicate wells, and both a blank control (only adding culture medium and CCK-8 reagent without seeding any cells) and a normal control group should be included (seed the same number of cells and add an equal volume of culture medium without adding any treatment drugs or reagents). 

### 2.15. Calcein AM/PI Detection of Apoptosis

BV-2 cells in the logarithmic phase were seeded into 96-well culture plates at a density of 5 × 10^4^ cells/mL and incubated overnight to allow cell adhesion. After adhesion, each well received 1 μM Aβ_1-42_ and 24 h incubation. After this incubation period, Calcein AM working solution was added to each well, followed by a 30 min incubation. A fluorescence microscope was used to observe the cells. Live cells exhibited green fluorescence, while dead cells exhibited red fluorescence. Each experimental condition was performed in triplicate to ensure the accuracy of the experimental data. Images were acquired using an Olympus fluorescence inverted microscope. A 20× objective lens was used, with the resolution set at 1024 × 768 pixels to ensure image quality and clarity.

### 2.16. Western Blotting

Target protein extraction: Cells were lysed using RIPA lysis buffer containing protease inhibitors and incubated on ice for 30 min to ensure complete protein extraction. The lysates were centrifuged at 13,400× *g* for 15 min at 4 °C, and the supernatant was collected to obtain total cellular proteins. Protein concentration in the samples was then quantified using a BCA protein assay kit. Finally, 5× SDS loading buffer was added to the samples, which were heated at 100 °C for 10 min to fully denature the proteins. After processing, the protein samples were stored frozen. SDS-PAGE was utilized for protein separation on a polyacrylamide gel with a concentration of between 6% and 20% sodium dodecyl sulfate. Afterwards, proteins were transferred to the PVDF membrane, which was then blocked using a closure solution. The membrane was co-incubated by the primary antibody for an overnight period at 4 °C. After the overnight incubation and rinsing with TBST, the appropriate anti-rabbit/mouse IgG antibody was selected based on the antibody genus and incubated for 1 h at room temperature. The protein bands were finally detected using enhanced chemiluminescence (ECL). We obtained the grayscale value of the protein bands using ImageJ software, then normalize the expression level of the target protein to the expression level of the internal control protein, and performed statistical analysis.

### 2.17. Quantitative Real-Time Fluorescent PCR (RT-PCR)

We prepared a suspension of mouse hippocampal tissue or BV2 cells in a 15 mL centrifuge tube, and centrifuged it at 175× *g* at room temperature for 15 min. The hippocampal tissue should be cut into small pieces and homogenized before use. After centrifugation, we removed the supernatant, carefully retained the pellet, and washed it three times with PBS. Then, we added Buffer RLT Plus lysis buffer for thorough lysis. For hippocampal tissue, after adding the lysis buffer, we transfered the mixture to a 1.5 mL centrifuge tube, and loaded the lysate onto a DNA clearance column and centrifuged it at 16,200× *g* for 1 min, retaining the filtrate. We then added 350 µL of 70% ethanol to the filtrate and mixed it immediately. Then, we loaded the lysate onto a DNA clearance column and centrifuged it at 16,200× *g* for 1 min, retaining the filtrate. We then added 700 µL of Buffer RW1 to the RNA binding column, letting it sit at room temperature for 30 s, and then centrifuged it at 16,200× *g* for 1 min, discarding the filtrate. We then added 500 µL of Buffer RW solution and repeated the 16,200× *g* centrifugation step twice. We then transferred the binding column to a 1.5 mL RNase-free centrifuge tube, added 40 µL of RNase-free water, and let it sit for 2 min, before centrifuging it at 16,200× *g* for 2 min to collect the RNA solution for cDNA synthesis. We then converted the RNA into cDNA, used the cDNA as a template, and perform the PCR according to the program settings in Table 1 and Table 2. Finally, the 2^−ΔΔCt^ method was adopted to analyze the levels of mRNA expression. 

### 2.18. Cell Transfection Assay

To complete this process, dilute the PepMute™ transfection buffer (5×) using sterile deionized water at a 1:4 ratio to prepare the working solution. Passage BV2 cells 18–24 h before transfection to ensure a cell density of approximately 50% at the time of transfection. Thirty minutes before transfection, add complete culture medium containing serum and antibiotics to the 6-well plate. Mix 10 pmol of siRNA per well (final concentration 10 nM) with 100 µL of the working solution, add 2.4 µL of PepMute™ Plus reagent, and incubate at room temperature for 15 min to form the transfection complex. Gently add the mixture dropwise onto the BV2 cells and return the plate to the incubator. After 24–48 h post-transfection, preliminarily assess transfection efficiency using fluorescence microscopy, followed by the detection of gene expression and protein level downregulation using qRT-PCR and Western blotting.

### 2.19. Preparation Procedure for Aβ_1-42_ Oligomers

To complete this process, equilibrate 1 mg of frozen Aβ_1-42_ dry powder at room temperature for 20 min. Centrifuge at 1200× *g* for 5 min to minimize wall adhesion and dosage loss. Add 220 μL of hexafluoroisopropanol (HFIP) to the centrifuged dry powder to dissolve the Aβ_1-42_. After dissolving, place the solution at room temperature for 1 h to ensure uniformity. Allow the dissolved Aβ_1-42_ solution to sit in a sterile fume hood for 4 h until the HFIP evaporates completely, leaving behind a white flocculent precipitate. Dissolve the Aβ_1-42_ peptide film on the wall of the test tube with 88 μL of dimethyl sulfoxide (DMSO) to prepare a 2.5 mM solution, and dilute it with serum-free high-glucose DMEM medium to a concentration of 400 μM. After incubation, centrifuge the Aβ_1-42_ solution at 2190× *g* for 10 min at 4 °C. Collect the supernatant, which contains the Aβ_1-42_ oligomers.

### 2.20. Statistical Analysis

In this study, the data were yielded from at least three independent experimental trials. GraphPad Prism 9 software ((La Jolla, CA, USA) was adopted for statistical analyses. Images of immunofluorescence staining were analyzed using ImageJ software. In the immunohistochemistry experiment, approximately 5 consecutive sections were stained for each mouse, with all sections taken from the same brain region. The staining results were quantitatively analyzed using ImageJ software. Positive staining areas were identified by setting appropriate thresholds to ensure automatic and consistent quantitative analysis. The Morris water maze escape latency and freezing time were analyzed using two-way ANOVA, while other experimental results were evaluated with one-way ANOVA. A *p*-value of less than 0.05 was considered as having statistical significance.

## 3. Results

### 3.1. TREM2 Is a Potential Target and Has Low Expression in AD

We started by collecting gene expression data from 30 AD patients, 7 with severe disease and 23 with non-severe disease, using the GEO dataset GSE28146 to elucidate TREM2′s role in AD progression (Figure 1A). We assessed and compared the relative expression levels of TREM2 between these two groups. The boxplot analysis revealed the significantly higher expression of TREM2 in non-severe samples than in severe samples. We acquired gene expression data from 87 AD patients from the GSE5281 dataset to investigate the role of *TREM2* in AD (Figure 1B). The patients were stratified into two groups based on the median *TREM2* gene expression level. The analysis then centered on identifying differences in *TREM2* mRNA expression between groups, applying criteria of |log2FC| > 1 and *p* < 0.05 to determine significantly differentially expressed genes. This analysis identified 63 differentially expressed genes (DEGs), which were visualized on a volcano plot. The findings, through GO and KEGG pathway enrichment, suggest that collagen-containing extracellular matrix and glutamatergic synapses are closely associated with the pathogenesis of AD (Figure 1C). Gene set enrichment analysis (GSEA) (Figure 1D) further demonstrated that samples with high TREM2 expression exhibited significant enrichment in the PI3K-Akt signaling pathway. In contrast, samples with low TREM2 expression showed significant enrichment in AD-associated pathways. To validate serum sTREM2 levels in clinical AD patients, we collected serum samples from 58 individuals, including 32 AD patients and 26 healthy controls. The analysis revealed a notable reduction in sTREM2 levels in the blood of AD patients compared with healthy controls. Serum sTREM2 levels in peripheral blood were measured using ELISA. The results (Figure 1E) showed a significant reduction in serum sTREM2 levels in the AD group (*p* < 0.001), consistent with previous findings. In the later stages of AD, TREM2 levels were markedly decreased, confirming that TREM2 is a reliable biomarker for AD.

### 3.2. Network Pharmacology Analyses of Potential Mechanisms of OUA Involvement in AD

Drawing from the identified active components of OUA and information from the AD database, we sought relevant potential targets to explore the mechanisms by which OUA treats AD. First, we identified 107 therapeutic targets of OUA. 

In this study, “Alzheimer’s disease” was used as the key search term to identify AD-related gene targets from the GeneCards, OMIM, and DrugBank databases. The gene targets obtained from these three databases were merged, and duplicate gene targets were removed, resulting in a total of 13,617 unique gene targets (Figure 2A). Ultimately, we intersected the therapeutic targets of OUA with the AD gene targets, resulting in 93 target genes (Figure 2B). GO enrichment analysis of the intersecting genes revealed that the biological processes involved include peptide–serine phosphorylation, reactive oxygen species metabolism, and the role of the Notch signaling pathway in regulating reactive oxygen species. The cellular components include the presynaptic membrane, early endosome, and membrane raft. The molecular functions involve serine/threonine–protein kinase activity (Figure 2D). KEGG pathway analysis results indicate that the potential anti-AD targets of OUA are enriched across 224 pathways. Excluding tumor-related pathways, the PI3K/AKT signaling pathway stood out with the highest number of enriched gene targets (Figure 2C). We constructed a PPI network diagram (Figure 2E) by varying the node size and color according to the degree. Among the interaction nodes with a degree above the median (≥12), 40 core target proteins were identified. Highly ranked core target proteins, including EGFR, STAT3, CASP3, PTGS2, ERBB2, ICAM1, GRB2, PIK3CA, CASP8, FYN, and SYK, are involved in inflammation and microglial activation, which are crucial for the treatment of AD with OUA. There are no reports available on whether it directly acts on the TREM2 protein to regulate microglial polarization. In this study, we performed the molecular docking of OUA with TREM2 (Figure 2H) and obtained the minimum binding energy. Generally, a lower binding energy between the ligand and receptor indicates better docking capability. A binding energy of <−5.0 kJ/mol (approximately −1.2 kcal/mol) is considered indicative of good binding affinity. The docking results show a binding energy of approximately −8.648 kcal/mol, which is less than −5 kcal/mol, indicating a strong binding affinity between OUA and the TREM2 molecule. Thus, our focus will be on elucidating how OUA targets TREM2 to exert its physiological effects.

### 3.3. The Effect of OUA on the Aβ_1-42_-Induced Change in BV-2 Cell Viability 

To assess the impacts of OUA and Aβ_1-42_ on BV-2 cell viability, cells were treated with varying concentrations of OUA (0 to 50 nM) for 24 h, then determined by a CCK-8 assay. The results (Figure 3B) indicate that OUA concentrations below 7.5 nM did not affect BV-2 cell viability. To identify the optimal concentration for intervention, BV-2 cells were treated with Aβ_1-42_ oligomers at doses ranging from 1 to 10 μM. The results (Figure 3C) indicate that at 1 μM, Aβ_1-42_ significantly decreased BV-2 cell viability by 23.08% (*p* < 0.0001). Subsequent experiments utilized an Aβ_1-42_ concentration of 1 μM. BV-2 cells were incubated with various concentrations of OUA for 24 h and analyzed using a CCK-8 assay. The results (Figure 3D) show that OUA mitigated the reduction in BV-2 cell viability caused by Aβ_1-42_. The morphological analysis (Figure 3E) revealed that resting-state microglia had a normal morphology, irregular shapes, highly branched processes, elongated or oval cell bodies, and strong adhesion. The nuclei were kidney-shaped, oval or triangular. After Aβ_1-42_ induction, BV-2 cells exhibited reduced adhesion, increased numbers of suspended cells, and an activated state, with oval cell bodies and retracted processes (Figure 3F). Compared to the Aβ_1-42_ group, the OUA-treated group exhibited increased numbers of microglial processes, and fewer suspended cells (Figure 3G). After co-culturing Aβ_1-42_-induced BV-2 cells with 7.5 nM OUA for 24 h, Calcein AM/PI staining was performed (Figure 3H), which showed that OUA attenuated Aβ_1-42_-induced cell damage, as evidenced by the reduction in red fluorescence. These experiments demonstrated that OUA could counteract the Aβ_1-42_-induced reduction in cell viability and improve cellular morphological changes.

### 3.4. OUA Increases Aβ_1-42_-Induced TREM2 Expression in BV-2 Cells and Modulates Microglial Polarization

iNOS serves as a marker for M1 polarization, whereas Arg-1 is indicative of M2 polarization [7]. We assessed the impact of OUA on the M1/M2 polarization of BV-2 cells by measuring iNOS and Arg-1 levels using Western blotting and immunofluorescence staining. We also investigated the neuroprotective effect of OUA in relation to BV-2 cell polarization. Additionally, Western blot analysis was utilized to assess alterations in TREM2 expression in BV-2 cells following induction by Aβ_1-42_ and subsequent treatment with different concentrations of OUA. The results (Figure 4A) show that 7.5 nM OUA significantly upregulated TREM2 expression, which prompted its use in subsequent experiments. The Western blotting analysis of protein expression in Aβ_1-42_-treated BV-2 cells revealed that Aβ_1-42_ treatment downregulated TREM2 protein expression, whereas OUA treatment upregulated it. Additionally, the results (Figure 4B,C) show that compared to the Aβ_1-42_ treatment, OUA treatment significantly decreased iNOS expression and elevated Arg-1 protein levels. The immunofluorescence analysis of iNOS and Arg-1 expression (Figure 4D,E) similarly showed that, compared to Aβ_1-42_ treatment, OUA treatment downregulated iNOS expression (weaker red fluorescence) and upregulated Arg-1 protein expression (stronger red fluorescence).

### 3.5. Effects of OUA on the Aβ_1-42_-Induced TREM2/PI3K/AKT Pathway

TREM2 signaling interacts with the adaptor proteins DNAX activation protein 12 (DAP12) and DAP10. DAP10 facilitates signal propagation through the recruitment of phosphoinositide 3-kinase (PI3K) [30]. PI3K activates the kinase AKT by mediating receptor tyrosine kinase signaling [31]. Activated AKT phosphorylates proteins that play roles in cell survival [32], the cell cycle, angiogenesis, and metabolism [33]. Western blotting was performed to assess the expression and phosphorylation levels of PI3K and AKT to elucidate OUA-regulated signaling pathways. The results (Figure 5A) show that the Aβ_1-42_-treated group exhibited reduced levels of p-PI3K and p-AKT without altering PI3K and AKT expression. OUA treatment notably enhanced p-PI3K and p-AKT levels. To further explore TREM2′s role in the PI3K/AKT signaling pathway, we knocked down TREM2 expression. As shown in Figure 5A,C,E, TREM2 silencing further decreased p-PI3K and p-AKT levels after Aβ_1-42_ treatment, while OUA treatment effectively reversed these reductions.

Additionally, compared to the NC group, the reduction in PI3K and AKT phosphorylation in the TREM2-silenced group was more pronounced. These results suggest that OUA enhances PI3K and AKT phosphorylation, while TREM2 silencing markedly diminishes it. OUA likely modulates microglial polarization through the PI3K/AKT pathway, thereby exerting its neuroprotective effects.

### 3.6. Effect of OUA on Aβ_1-42_ -Induced Inflammatory Cytokine Production in BV-2 Cells

We conducted RT-PCR to determine whether OUA modulates the expression levels of inflammatory cytokines in BV-2 cells. The results (Figure 5F,G) indicate that after Aβ_1-42_ induction, the level of proinflammatory cytokine IL-1β production was significantly elevated compared to the NC group, while IL-4 production was significantly decreased. In the OUA treatment group, the Aβ_1-42_-induced synthesis of the proinflammatory cytokine IL-1β was inhibited, whereas IL-4 levels increased. When TREM2 was silenced, Aβ_1-42_-induced IL-1β production was significantly elevated compared to the NC group, while IL-4 production significantly decreased (Figure 5H,I). These findings suggest that OUA exerts neuroprotective effects by upregulating anti-inflammatory IL-4 and suppressing proinflammatory IL-1β in Aβ_1-42_-treated BV-2 cells. TREM2 plays a pivotal role in this process.

### 3.7. OUA Treatment Improves Cognitive Dysfunction in FAD^4T^ Mice

The Morris Water Maze and fear conditioning test are classic cognitive models [34]. We evaluated the spatial learning and memory capabilities of mice using the aforementioned behavioral methods. Results from the place navigation test in the MWM indicate that, starting on the fourth day, the delay in finding the target platform was significantly longer in FAD^4T^ mice compared to WT mice (Figure 6B, *p* < 0.01), indicating spatial learning and memory deficits in the FAD^4T^ mice. Compared with the FAD^4T^ group, the group injected intraperitoneally with OUA showed a significantly shorter delay in reaching the target platform (Figure 6C, *p* < 0.05). The spatial exploration test results on the fifth day indicate that FAD^4T^ mice crossed the original platform quadrant significantly fewer times (Figure 6D, *p* < 0.001). OUA significantly increased the swimming distance (Figure 6F, *p* < 0.01) and the time spent in the target quadrant by FAD^4T^ mice (Figure 6E, *p* < 0.01). These findings suggest that OUA markedly improves spatial learning and memory in FAD^4T^ mice.

In the fear conditioning memory experiment, the third day’s freezing time represented contextual memory, while the fourth day’s freezing time represented cued fear memory. The results show that there was no significant difference in freezing time between the experimental groups during the acclimatization period (Figure 6G,H, *p* > 0.05). Compared to the adaptation period, WT mice showed a significant increase in freezing time during the contextual and cued fear memory test periods (Figure 6G,H, *p* < 0.05), indicating that they formed contextual and cued fear memories after receiving shock stimuli. However, the freezing time of FAD^4T^ mice did not significantly increase during the contextual and cued fear memory test periods (Figure 6G,H, *p* > 0.05), suggesting that FAD^4T^ mice had impaired fear memory acquisition. OUA treatment improved these deficits. OUA treatment markedly extended the freezing time in FAD^4T^ mice during both contextual and cued fear memory tests (Figure 6E,I,J, *p* <0.05). These findings suggest that OUA significantly improves fear memory in FAD^4T^ mice.

### 3.8. OUA Alleviates Neuronal Damage and Pathological Changes in the Hippocampal Region of FAD^4T^ Mice

In this study, we observed neuronal damage and pathological changes in different groups of mice using HE and Nissl staining. HE staining results (Figure 7B) indicated that the WT group displayed no significant pathological changes in the hippocampus, with densely packed pyramidal cells, normal morphology, a large number of cells, normal staining, uniform distribution, orderly arrangement, and intact structure without necrotic or degenerated neurons. In the FAD^4T^ group, the hippocampal neurons were significantly reduced in number and exhibited abnormal cell morphology, reduced cell volume, shrunken nuclei, deeper staining, disordered arrangement, and loose structure. Compared to the AD model group, the OUA group showed significant improvement in hippocampal pathology, characterized by an increased number of cells, more intact structures, a more orderly arrangement, and fewer abnormal cells.

Nissl staining is used to assess neuronal damage and loss [35]. In this study, Nissl staining was utilized to quantify neurons in the hippocampal CA1, CA3, and cortical regions. This method was also employed to morphologically evaluate neuronal changes in the hippocampus and cortex of mice following OUA treatment. Nissl staining (Figure 7C) showed densely packed pyramidal cells in the CA1, CA3, and cortical areas, with intact structures and abundant dark blue Nissl bodies in the cytoplasm. In the FAD^4T^ group, neurons located in the hippocampal CA1, CA3, and cortical areas were irregularly arranged, triangular or polygonal in shape, fewer in number, and had significantly reduced or no Nissl bodies in the cytoplasm. The results from HE and Nissl staining indicate that OUA treatment reduced neuronal damage and loss in both the hippocampus and cortex. These findings suggest that OUA may mitigate neuronal death in FAD^4T^ mice.

### 3.9. OUA Alleviates Synaptic Toxicity in the Hippocampus of FAD^4T^ Mice

Synapse formation was studied using immunofluorescence labeling to observe changes in SYN and PSD95 expression in the mouse hippocampus. As shown in Figure 7D, numerous SYN-positive cell bodies and neurites were abundant and regularly arranged in the hippocampal region of WT mice. Conversely, FAD^4T^ mice exhibited a significant decrease in SYN-positive neurons and neurites, noticeable neurite damage, and disordered dendritic arrangements. Compared with the FAD^4T^ group, the OUA intervention group showed an increase in SYN-positive cells and a significant enhancement in both neurite number and length. Immunofluorescence labeling for PSD95 (Figure 7E) revealed a decrease in PSD95-positive cells in the hippocampal neurons of FAD^4T^ mice compared to WT mice. However, the OUA-treated group exhibited a significant increase in PSD95-positive cells compared to the FAD^4T^ group. Notably, PSD95 protein downregulation was more pronounced than SYN in FAD^4T^ mice, suggesting that postsynaptic receptor damage might play a more critical role in AD pathogenesis.

### 3.10. OUA Alleviates the Proliferation and Activation of Astrocytes and Oligodendrocytes in the Brains of FAD^4T^ Mice

In this study, we employed GFAP and OLIG2 antibodies for immunohistochemical staining to evaluate the effects of OUA on astrocytes and oligodendrocytes. The results (Figure 8A) indicate fewer activated GFAP-positive astrocytes in the CA1 and CA3 regions of the hippocampus in WT mice, while clear activation was observed in the same regions of FAD^4T^ mice. In contrast, FAD^4T^ mice treated with OUA showed a significant reduction in activated astrocytes in the CA1 and DG areas of the hippocampus compared to the model group. Likewise, immunohistochemical staining with OLIG2 antibody (Figure 8B) revealed fewer activated oligodendrocytes in the corpus callosum (CC) and cingulate gyrus (CG) regions of WT mice. Conversely, FAD^4T^ mice showed a marked increase in activated oligodendrocytes in the CC and CG regions, an effect that was effectively reversed by OUA treatment.

### 3.11. OUA Regulates Microglial Cell Polarization Through the TREM2/PI3K/AKT Pathway in FAD^4T^ Mice 

To investigate the impact of OUA on microglial phenotype changes and channel protein expression, we analyzed TREM2 and related proteins using Western blotting. The Western blotting results (Figure 7E,F) indicate that OUA upregulated Arg-1 expression and downregulated iNOS expression, thereby modulating the polarization phenotype of microglia in FAD^4T^ mice. TREM2 protein levels were significantly reduced in the hippocampus of FAD^4T^ mice, and 2 µg/kg OUA reversed this reduction. By activating p-PI3K and p-AKT, OUA facilitated the transition of microglia to the M2 phenotype, exerting positive anti-AD effects. These findings suggest that OUA’s physiological effects involve modulating microglial polarization and enhancing anti-AD activity.

### 3.12. OUA Promotes TREM2 Expression and Inhibits Neuroinflammation in FAD^4T^ Mice

Inflammatory cytokine expression in mice was measured using qRT-PCR. The results (Figure 8G–J) indicate that FAD^4T^ mice had increased transcript levels of the proinflammatory cytokines IL-1β and TNF-α, and decreased transcript levels of the anti-inflammatory cytokines IL-4 and IL-10, compared with WT mice. In the OUA-treated group, the transcript levels of TNF-α and IL-1β were reduced in FAD^4T^ mice, while the production of the anti-inflammatory cytokines IL-4 and IL-10 was increased. These findings suggest that OUA exerts neuroprotective effects in FAD^4T^ mice by upregulating anti-inflammatory cytokines (IL-4 and IL-10) and downregulating proinflammatory cytokines (IL-1β and TNF-α).

## 4. Discussion

This study investigates the therapeutic role of OUA in improving learning and memory deficits associated with AD, and elucidates the underlying molecular mechanisms in detail. Our results indicate that OUA improves cognitive deficits in FAD^4T^ mice and modulates microglial M1/M2 polarization both in vivo and in vitro. Furthermore, through bioinformatics analysis, molecular docking, and experimental validation, we identified TREM2 as a potential target of OUA. TREM2 regulates PI3K/AKT pathway activation, playing a critical role in OUA-mediated M2 microglial polarization and its anti-inflammatory effects on Aβ_1-42_-induced BV-2 cells and FAD^4T^ mice.

Over the past few decades, researchers have developed drugs based on the amyloid hypothesis to eliminate Aβ plaques, inhibit Aβ aggregation and deposition, or reduce its brain deposition by inhibiting γ-secretase and β-secretase. However, these therapeutic strategies have all failed in clinical trials [36]. The failure of anti-Aβ immunotherapy clinical trials has shifted researchers’ focus to other AD-related mechanisms, such as neuroinflammation. Recent large-scale exome sequencing studies have identified many AD susceptibility genes in microglia, suggesting a strong link between brain immune dysfunction and the pathological progression of AD [28]. OUA is a Na+/K+-ATPase (sodium pump) inhibitor, and regulates physiological processes such as cell proliferation and apoptosis through Na^+^/K^+^-ATPase signaling pathways [37]. Many studies have shown that ouabain has toxic effects, such as inducing apoptosis proinflammatory effects at high doses [38,39,40], whereas low doses of OUA have beneficial effects on alleviating various diseases, such as myocardial ischaemia, tumors, and retinal damage [26,41,42]. OUA modulates various immune functions, including inflammation, in a dose-dependent manner by acting on Na^+^/K^+^ pump binding. Studies have shown that low concentrations of ouabain significantly reduce LPS-induced neuroinflammatory factors including IL-1, TNFα, and IKBKE [43]. Ouabain targets the transcription factor EB (TFEB), where enhanced TFEB activity increases lysosomal degradation pathways, leading to the degradation of APP in AD mice [44]. Therefore, OUA has shown promising anti-AD effects in early-stage research.

The activation and phenotypic modulation of microglia are critical for neuroprotection, and their activation leading to a neuroinflammatory response is a central pathological feature of several neurodegenerative diseases [45,46]. More than 30 AD risk loci have been identified by global genome analysis or whole-exome sequencing studies. Most of the identified AD-related genes are highly or exclusively expressed in microglia, thereby deepening our understanding of AD pathogenesis [28,47]. Among these susceptibility genes are ApoE and TREM2, the most significant risk genes for sporadic AD. Microglia can be categorized into M1 and M2 types according to their activation mechanisms. M1 microglia primarily secrete proinflammatory factors, and their prolonged activation is considered cytotoxic to neurons, whereas M2 microglia have phagocytic capabilities and promote neuronal growth, which are considered beneficial to neurons [48,49,50]. Yun Xu and colleagues, through GWASs and single-cell transcriptomic studies, revealed that OUA may be significant in treating AD [28]. The study found that OUA significantly increased TREM2 protein expression in BV-2 cells. In Aβ_1-42_ -induced BV-2 cells, TREM2 expression was downregulated, whereas OUA pretreatment upregulated TREM2 and M2 marker expression. TREM2 knockdown diminished OUA’s impact on M2 microglial polarization and inflammation resolution. Animal experiments using the Morris Water Maze and fear conditioning tests indicated that OUA improved spatial learning and memory abilities in FAD^4T^ mice. Additionally, HE and Nissl staining showed that OUA significantly alleviated neuronal damage in AD mice. These findings confirm that OUA significantly ameliorates AD-related neuropathology and improves cognitive function in FAD^4T^ mice, as evidenced by behavioral tests and histological analysis. 

TREM2 can bind to known ligands such as lipopolysaccharide (LPS), phospholipids, lipoproteins, and Aβ, among others [30,51,52,53]. Upon ligand binding, the transmembrane domain of TREM2 interacts with the DAP12 protein, which includes an immunoreceptor tyrosine-based activation motif (ITAM), via oppositely charged amino acid residues. This interaction transmits signals into the cell, resulting in the phosphorylation of tyrosine residues in the ITAM region and the recruitment of spleen tyrosine kinase (SYK). SYK activation triggers a signaling cascade that activates phosphatidylinositol 3-kinase (PI3K), leading to calcium ion activation, integrin activation, and cytoskeletal rearrangement, and it can modulate the mammalian target of rapamycin (mTOR) and mitogen-activated protein kinase (MAPK) signaling. These processes influence various physiological functions of microglia, including metabolism, proliferation, cytokine release, phagocytosis, and migration [54,55,56,57]. Synaptophysin (SYN) is a calcium-binding glycoprotein found in all axon terminals and is specifically located on the presynaptic vesicle membrane. SYN is involved in channel formation, synaptic vesicle recycling, and neurotransmitter release. SYN is recognized as a crucial marker of synaptogenesis and synaptic remodeling [58]. Postsynaptic density protein-95 (PSD95) serves as the main scaffold protein on the postsynaptic membrane. It is essential for the activity and stability of postsynaptic membrane receptors, regulates the synapse number during development, promotes synapse formation, and is closely related to synaptic plasticity and learning and memory regulation in individuals with AD [59,60]. The SYN and PSD95 proteins are markers of synaptic reconstruction, and effectively reflect the compensatory regeneration of neurons and axons. In this study, the expression levels of SYN and PSD95 proteins in the mouse hippocampus were examined by immunofluorescence staining. The results indicate a significant reduction in SYN and PSD95 protein expression in the hippocampus of mice, while the OUA intervention group ameliorated the pathological damage in the hippocampus of the FAD^4T^ group. The experimental results indicate that SYN and PSD95 are the main targets of synaptic structural damage in the hippocampal region of FAD^4T^ mice, with more pronounced damage to postsynaptic structures. The reduced expression of SYN and PSD95 proteins destroys synaptic plasticity, resulting in significant reductions in learning and memory. These experimental results suggest that a key mechanism by which OUA improves cognitive dysfunction is promoting synapse formation and synaptic transmission to repair synaptic structural damage in the hippocampus. Astrocytes and oligodendrocytes also play crucial roles in neuroinflammation. This study examined the expression of the astrocyte and oligodendrocyte marker proteins GFAP and OLIG2 using immunohistochemistry. In FAD^4T^ mice, the CA1 and CA3 regions showed elevated GFAP expression and notable astrocyte activation, while the CC and CG regions displayed increased OLIG2 protein levels. On the other hand, the OUA inhibited astrocyte and oligodendrocyte activation in these brain regions. Taken together, these results suggest that OUA improves AD-related neurological features and cognitive deficits in FAD^4T^ mice.

In this study, OUA increased TREM2 expression in both BV-2 cells and the hippocampus of FAD^4T^ mice, modulated p-PI3K and p-AKT levels in Aβ_1-42_-induced BV-2 cells, and enhanced the morphology and viability of these cells. In animal experiments, the MWM test and fear conditioning test showed that OUA enhanced the spatial learning and memory capabilities of FAD^4T^ mice. Additionally, OUA diminished hippocampal inflammation and enhanced the production of anti-inflammatory factors. HE and Nissl staining indicated that OUA significantly alleviated neuronal damage in FAD^4T^ mice. These findings confirm that OUA mitigates AD-related neuropathology and cognitive deficits in FAD^4T^ mice. BV-2 cells received various concentrations of OUA treatment. The results show that 7.5 nM OUA significantly upregulated TREM2 expression in BV-2 cells, leading us to select 7.5 nM OUA as the optimal concentration for subsequent experiments. We then verified whether the regulatory effect of OUA on microglial polarization and anti-inflammatory functions involved the PI3K/AKT signaling pathway activation. In BV-2 cells induced by Aβ_1-42_, TREM2 expression was reduced; however, pretreatment with OUA increased the levels of TREM2, p-PI3K, and p-AKT. In comparison to untreated FAD^4T^ mice, OUA treatment elevated TREM2 protein expression and also upregulated p-PI3K and p-AKT protein expression levels. OUA significantly enhanced the PI3K/AKT signaling pathway activity in both in vivo and in vitro models. Blocking the TREM2 signaling pathway decreased the activity of downstream proteins p-PI3K and p-AKT, thereby reducing OUA’s anti-inflammatory effects. We propose that OUA influences microglial polarization via the TREM2/PI3K/AKT pathway, lowering inflammatory mediator expression and providing neuroprotection. In this study, the concentration of ouabain we used was based on previous cell experiments and animal model studies [61,62]. Although these concentrations approach the toxicity threshold, they do not exceed it. Compared to plasma levels in patients, the concentrations used in in vitro cell experiments are higher in order to simulate a stronger pharmacological effect, though this does not fully reflect the actual plasma levels in clinical settings. Ouabain is a cardiac glycoside, a class of drugs associated with a high risk of serious side effects such as visual disturbances and arrhythmias. While ouabain is generally considered more toxic than digoxin, during our experiments, the mice did not exhibit significant side effects such as abnormal behavior or arrhythmias. Considering the toxicity and potential side effects of the drug on elderly AD patients, and given that elderly patients may be more sensitive to drug toxicity, the further optimization of ouabain’s dosage will be needed in future clinical applications. Moreover, additional studies should be conducted to ensure its safety.

## 5. Conclusions

In conclusion, OUA modulates microglial polarization, exerts anti-inflammatory effects, and its anti-AD properties are mediated through activation of the TREM2/PI3K/AKT pathway. Our research highlights the significant potential of TREM2 activation in suppressing neuroinflammation and combating AD. Targeting the TREM2/PI3K/AKT pathway to reduce neuroinflammation offers a promising strategy for treating AD and other neuroinflammatory disorders.

## Figures and Tables

**Figure 1 nutrients-16-03558-f001:**
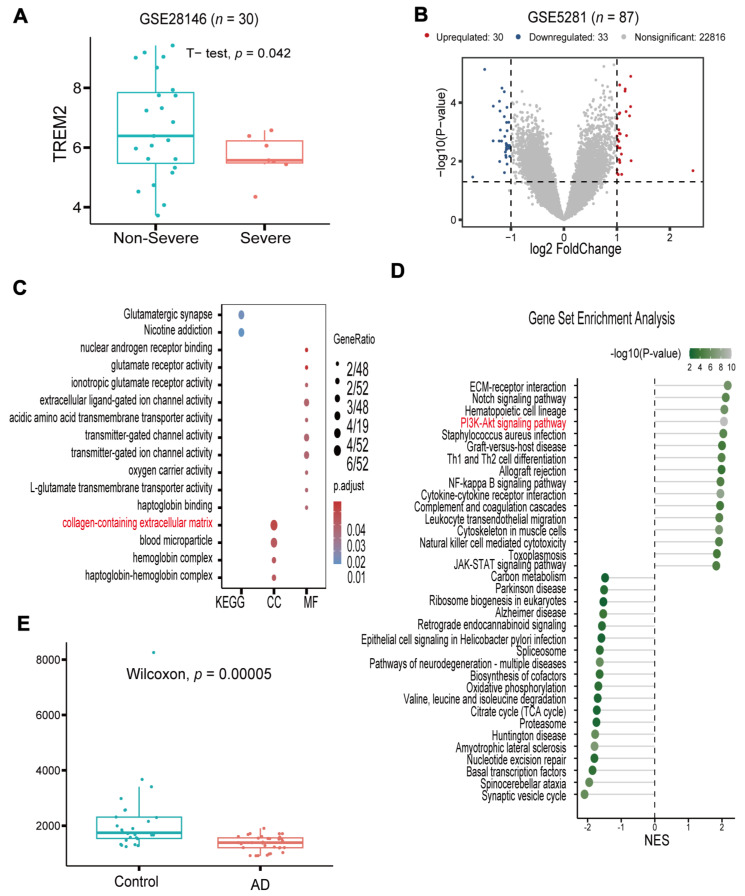
The *TREM2* gene is associated with AD progression. (**A**) Boxplot showing the expression level of *TREM2* in samples from patients with severe AD and non-severe AD (GSE28146). (**B**) Volcano plot of differentially expressed genes between samples with high and low *TREM2* expression levels (GSE5281). (**C**) GO and KEGG analyses of the differentially expression genes. (**D**) Lollipop plot of GSEA results for the differentially expressed genes. The PI3K-AKTsignaling pathway is marked in red. (**E**) Boxplot showing the relative abundance of *TREM2* in clinical AD and control samples.

**Figure 2 nutrients-16-03558-f002:**
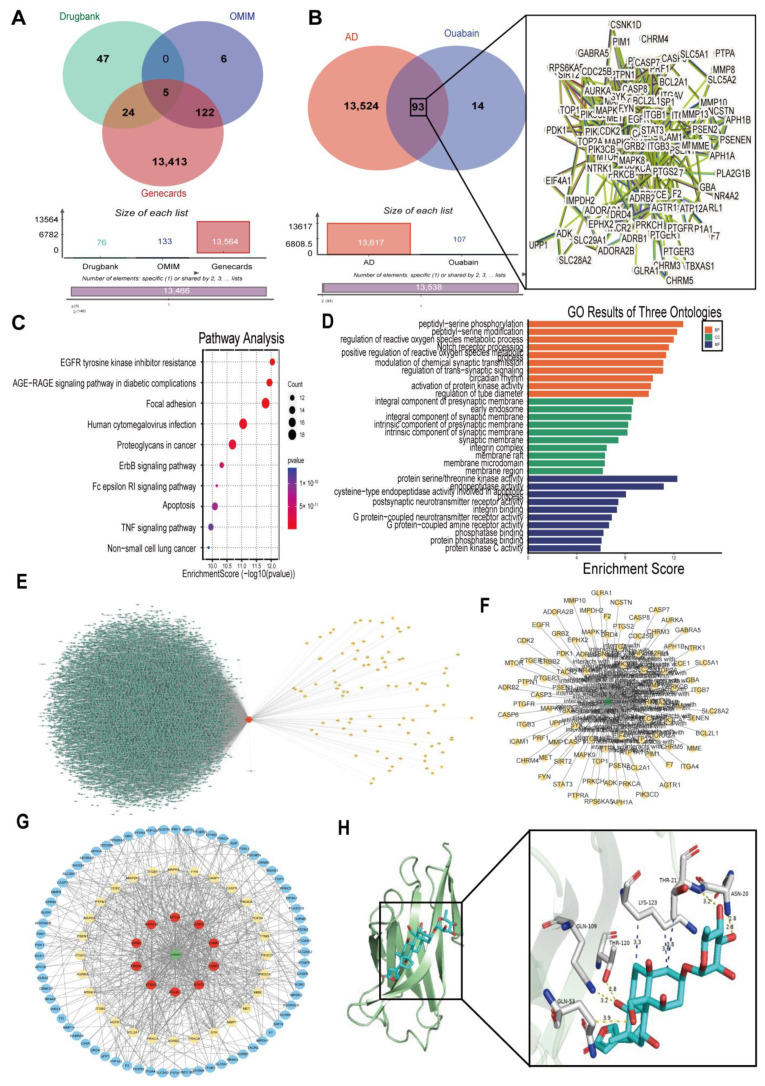
Ouabain network pharmacology and molecular docking. (**A**) Venn diagram showing the overlap of AD-related targets obtained from various databases. (**B**) Venn diagram of the overlap between AD-related targets and potential targets of OUA and the ouabain–AD target PPI network. (**C**) Predicted target signaling pathway enrichment diagram. (**D**) Results of the GO enrichment analysis. (**E**) Ouabain–target network. (**F**,**G**) Ouabain–AD target network. (**H**) Molecular docking model of ouabain–TREM2.

**Figure 3 nutrients-16-03558-f003:**
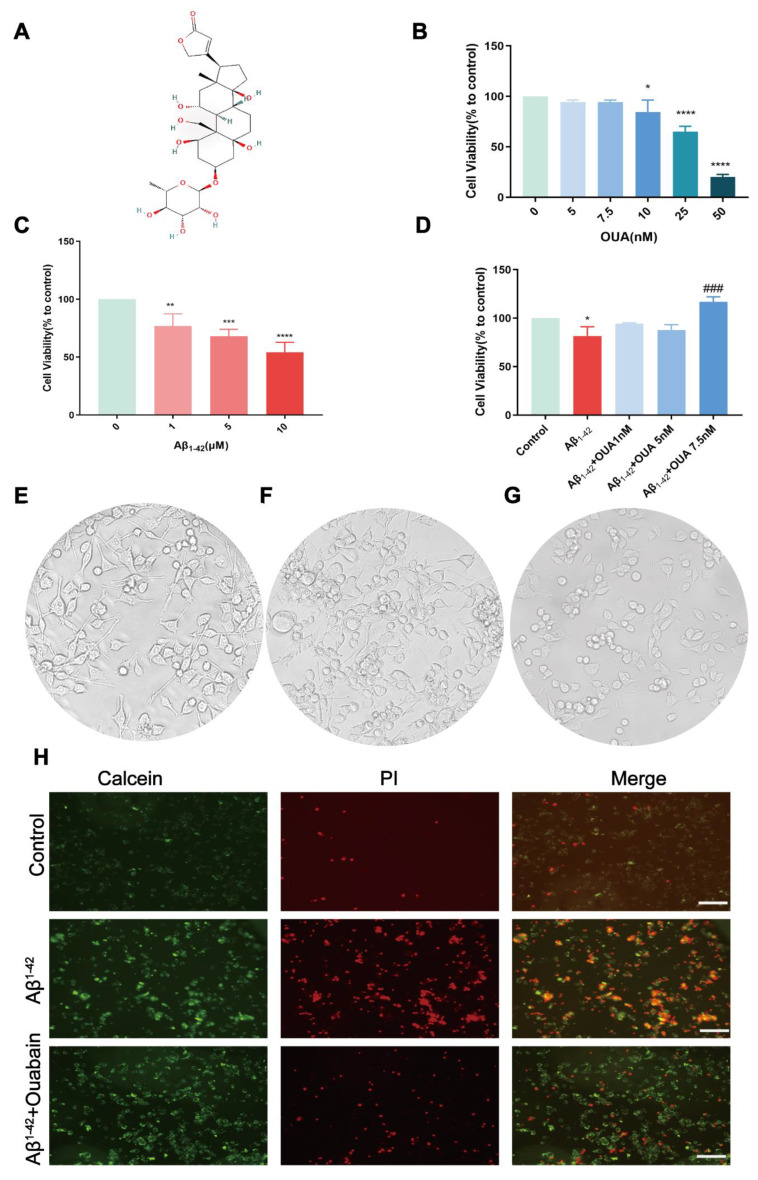
Effect on BV2 cell viability with and without Aβ_1-42_ induction. (**A**) Chemical structure of OUA. (**B**) Effects of different concentrations of OUA on cell viability. (**C**) Effects of different concentrations of Aβ_1-42_ on cell viability. (**D**) Effect of OUA on Aβ_1-42_-induced cell viability. Cell morphology of the following groups observed under an optical microscope (200×): (**E**) control, (**F**) model group (Aβ_1-42_), and (**G**) OUA+Aβ_1-42_ group. (**H**) Calcein–AM/PI staining of cells treated with Aβ_1-42_ and OUA. The data are presented as the means ± S.E.M.s, n = 3. * *p* < 0.05, ** *p* < 0.01, *** *p* < 0.001 and **** *p* < 0.0001 compared to the control group; ^###^
*p* < 0.001 compared to the model group. Scale bar = 50 μm.

**Figure 4 nutrients-16-03558-f004:**
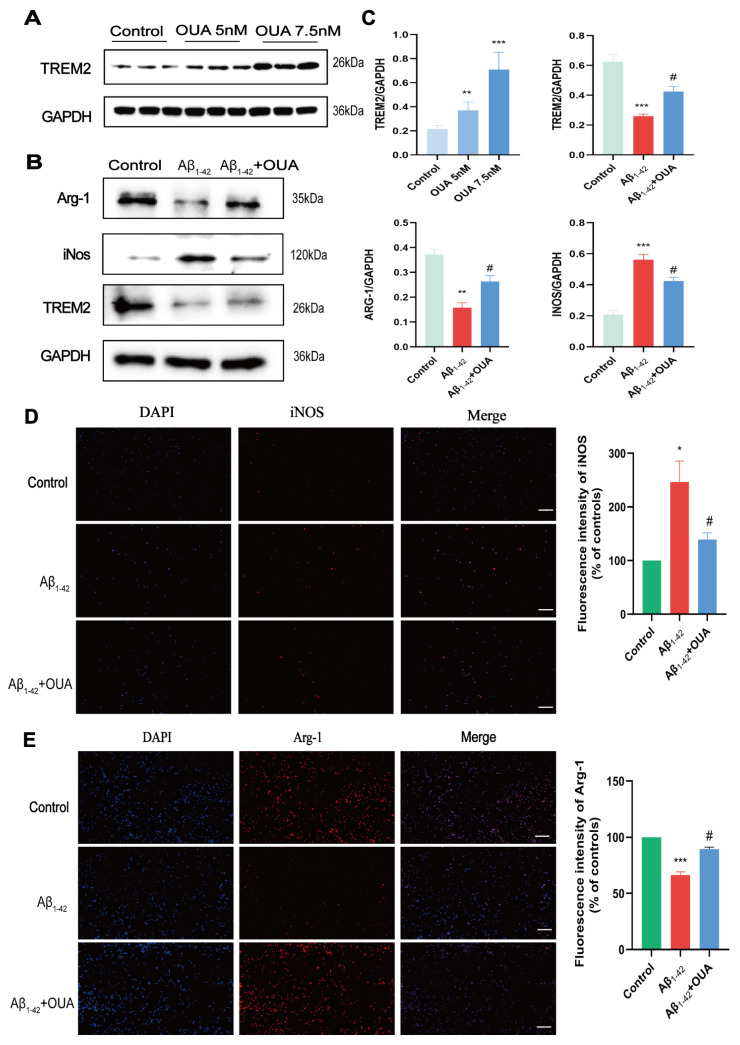
Effect of OUA on the expressions of TREM2, Arg-1, and iNOS in BV-2 cells. (**A**,**B**) Representative TREM2, Arg-1, and iNOS protein levels determined using Western blot analysis. (**C**) The levels of TREM2, Arg-1, and iNOS. (**D**) Immunofluorescence detection of iNOS expression (100×); scale bar = 100 μm. (**E**) Immunofluorescence detection of Arg-1 expression (100×). The blue fluorescence in the figure represents DAPI staining, while the red fluorescence indicates target protein staining. The data are presented as the means ± S.E.M.s, n = 3. * *p* < 0.05, ** *p* < 0.01, and *** *p* < 0.001 compared to the control group; ^#^
*p* < 0.05 compared to the model group. Scale bar = 100 μm.

**Figure 5 nutrients-16-03558-f005:**
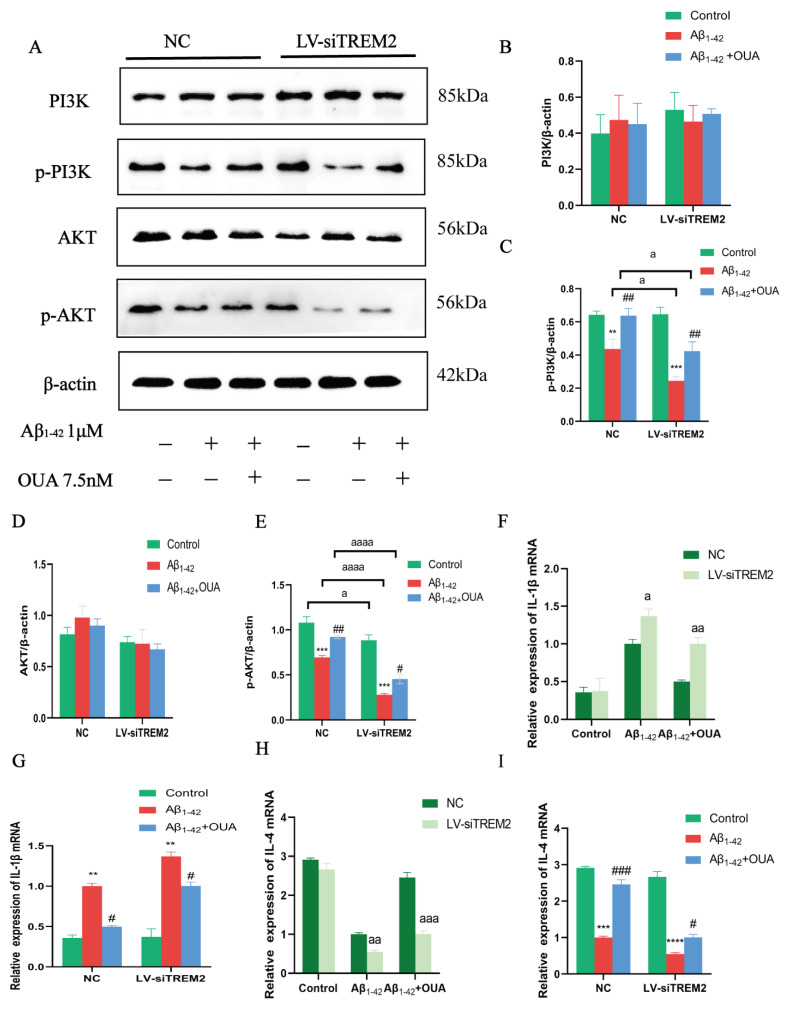
Effects of OUA on the expression of channel-associated proteins and inflammatory factors. (**A**) Representative Western blot analysis of the levels of the PI3K, p-PI3K, AKT, and p-AKT proteins. (**B**–**E**) The levels of PI3K, p-PI3K, AKT, and p-AKT. The expression of IL-1β (**H**,**I**) and IL-4 (**F**,**G**) was detected using qPCR. The data are presented as the means ± S.E.M.s, n = 3. ** *p* < 0.01, *** *p* < 0.001 and **** *p* < 0.0001 compared to the corresponding control group; ^#^
*p* < 0.05, ^##^
*p* < 0.01 and ^###^
*p* < 0.001 compared to the corresponding model group; ^a^
*p* < 0.05, ^aa^
*p* < 0.01, ^aaa^
*p* < 0.001 and ^aaaa^
*p* < 0.0001 compared to the NC group under the same experimental conditions.

**Figure 6 nutrients-16-03558-f006:**
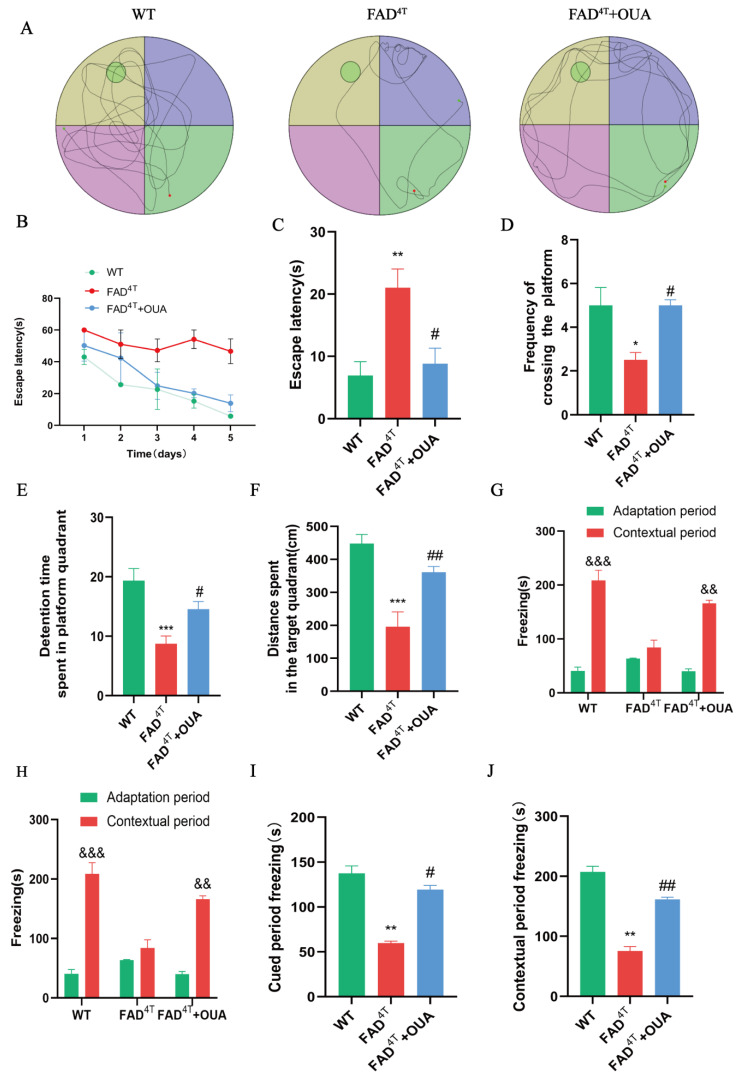
OUA improves spatial learning and memory abilities in FAD^4T^ mice. (**A**) Representative trajectory plots in the spatial exploration test. The four colors in the figure represent different quadrants. (**B**) Escape latency in the place navigation test. (**C**) Escape latency in the spatial exploration test. (**D**) Frequency at which mice crossed the original platform location in the experiment. (**E**) Time spent in the target quadrant. (**F**) Swimming distance in the target quadrant. (**G**) Freezing time during the adaptation period and the contextual fear memory test. (**H**) Freezing time during the adaptation period and the cued fear memory test. (**I**) Freezing time during the contextual fear memory test. (**J**) Freezing time during the cued fear memory test. The data are presented as the means ± S.E.M.s, n = 6. * *p* < 0.05, ** *p* < 0.01, and *** *p* < 0.001 compared to the WT group; ^#^
*p* < 0.05 and ^##^
*p* < 0.01 compared to the FAD^4T^ group; ^&&^
*p* < 0.01, and ^&&&^
*p* < 0.001 when comparing the contextual or cued fear memory test periods to the adaptation period.

**Figure 7 nutrients-16-03558-f007:**
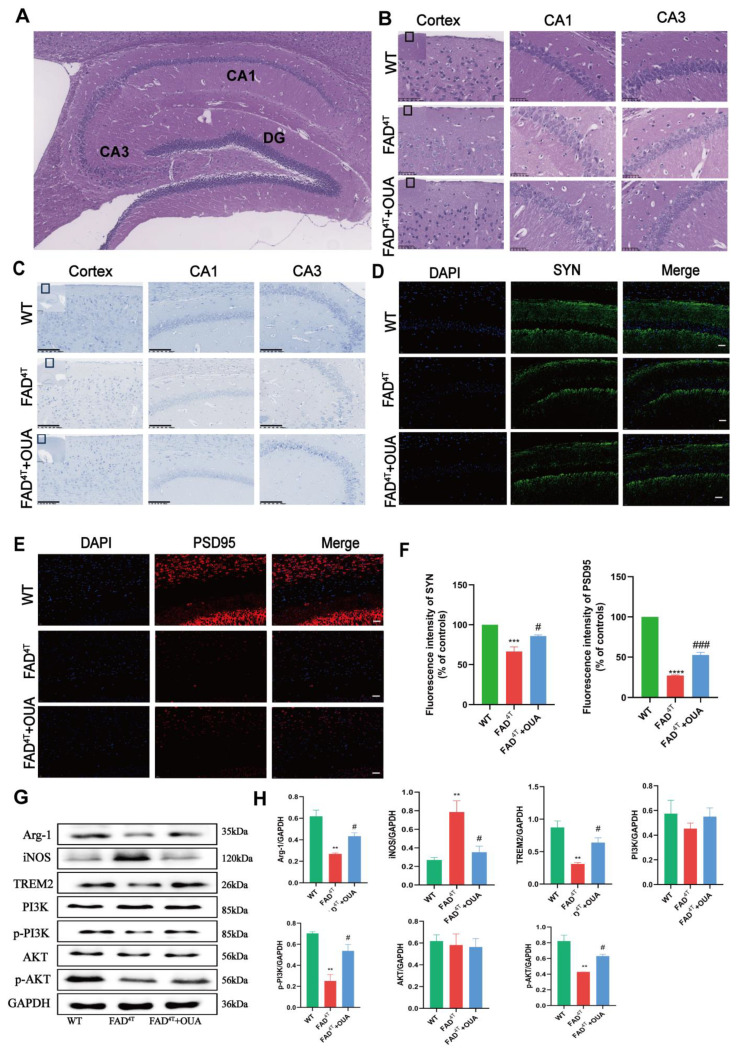
OUA attenuates neuronal and synaptic damage and pathological changes and effects on signaling pathways in FAD^4T^ mice. (**A**) Schematic diagram of mouse hippocampal structure. (**B**) HE staining of the hippocampus (200×); scale bar = 100 μm. (**C**) Nissl staining of the hippocampus (200×); scale bar = 100 μm. (**D**) Immunofluorescence detection of SYN expression (400×); scale bar = 20 μm. The black boxes in the figure indicate areas of magnification or focus. (**E**) Immunofluorescence detection of PSD98 expression (400×); scale bar = 20 μm. (**F**) Quantitative analysis of SYN and PSD95 fluorescence intensity. (**G**) Representative Arg-1, iNOS, TREM2, PI3K, p-PI3K, AKT, and p-AKT protein levels determined using Western blot analysis. (**H**) The levels of Arg-1, iNOS, TREM2, PI3K, p-PI3K, AKT, and p-AKT. The data are presented as the means ± S.E.M.s, n = 3. ** *p* < 0.01, *** *p* < 0.001 and **** *p* < 0.0001 compared with the corresponding control group; ^#^
*p* < 0.05 and ^###^
*p* < 0.001 compared with the corresponding model group.

**Figure 8 nutrients-16-03558-f008:**
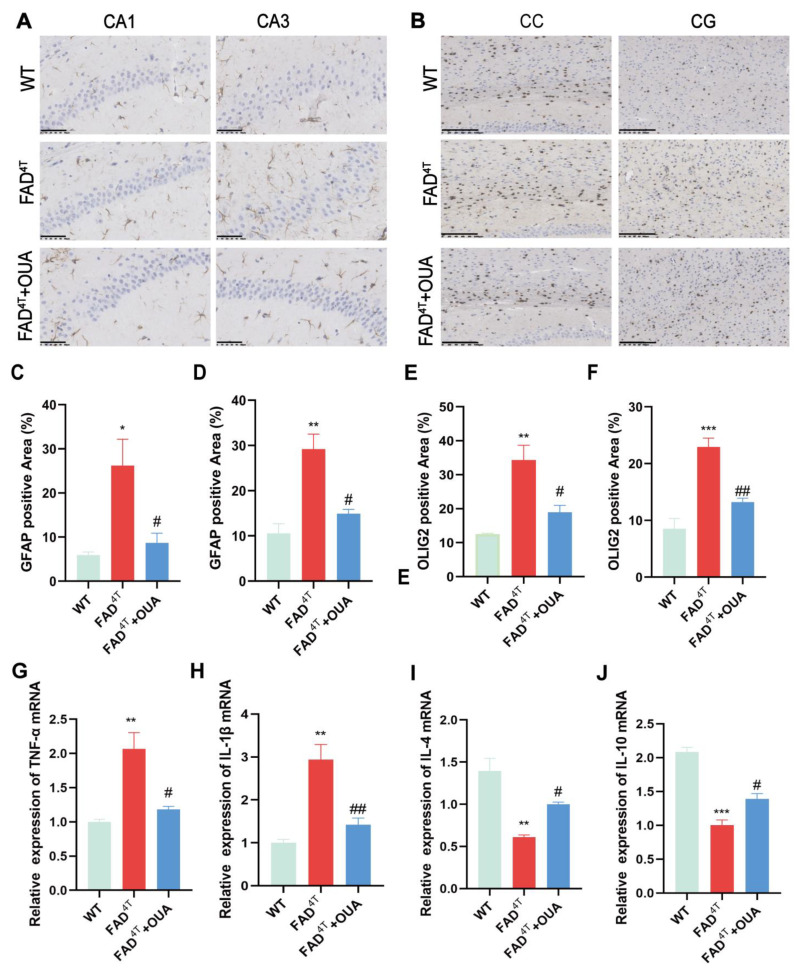
OUA attenuates the proliferation and activation of astrocytes and oligodendrocytes and the effects on inflammatory factors in the brains of FAD^4T^ mice. (**A**) Immunohistochemical detection of GFAP expression in the hippocampus (200×). Scale bar = 50 μm. (**B**) Immunohistochemical detection of OLIG2 expression in the CC and CG (200×). Scale bar = 100 μm. (**C**,**D**) Statistical map of positive areas in the CA1 and CA3 regions of the hippocampus. (**E**,**F**) Statistical map of positive areas in the CC and CG regions. The expression of TNF-α (**G**), IL-1β (**H**), IL-4 (**I**), and IL-10 (**J**) was detected using qPCR. The data are presented as the means ± S.E.M.s, n = 3. * *p* < 0.05, ** *p* < 0.01, and *** *p* < 0.001 compared with the corresponding control group; ^#^
*p* < 0.05 and ^##^
*p* < 0.001 compared with the corresponding model group.

**Table 1 nutrients-16-03558-t001:** Primers of qRT-PCR.

Gene Name	Forward Primer (5′->3′)	Reverse Primer (5′->3′)
*TREM2*	GCCTTCCTGAAGAAGCGGAA	GAGTGATGGTGACGGTTCC
*TNF-α*	CTGAACTTCGGGGTGATCGG	GGCTTGTCACTCGAATTTTGAGA
*IL-1β*	GAAATGCCACCTTTTGACAGTG	TGGATGCTCTCATCAGGACAG
*IL-4*	GGTCTCAACCCCCAGCTAGT	GCCGATGATCTCTCTCAAGTGAT
*IL-10*	CTTACTGACTGGCATGAGGATCA	GCAGCTCTAGGAGCATGTGG
*GAPDH*	AAGAAGGTGGTGAAGCAGG	GAAGGTGGAAGAGTGGGAGT

**Table 2 nutrients-16-03558-t002:** qPCR reaction system.

Component	Volumetric (20 μL)
cDNA	2 μL
Forward primer	0.4 μL
Universal miRNA qPCR Primer	0.4 μL
PerfectStart^TM^ Green qPCR SuperMix	10 μL
RNAase free Water	Make up to 20 μL

## Data Availability

The data supporting the results of this study are available from the corresponding author upon request.

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
