# Peer review of "Ouabain Ameliorates Alzheimer’s Disease-Associated Neuropathology and Cognitive Impairment in FAD4T Mice"

_nutrients, 2024, doi:10.3390/nu16203558_

Round 1
Reviewer 1 Report
Comments and Suggestions for Authors
In this paper, the authors provide substantial evidence to support a beneficial role of ouabain, a cardiac glycoside, in the treatment of the cognitive deficits of Alzheimer's disease.
The multidisciplinary nature of results strengthen the paper and the results are of potential interest.
However, I have several areas of concern and ask the authors to consider the questions below.
Please clarify food, water and light schedule for the mice.
Please provide RRIDs for all antibodies, if possible.
Please clarify the method used to protect against false positives in Bioinformatics analyses.
GO and KEGG analysis
Please clarify how this was done - were all proteins expected to interact with oubain and that are involved with AD simply placed in a list and uploaded? Please provide more detail.
Are enrichment results similar when using DAVID?
Animal groups and drug administration
Please provide power analyses and sample size calculations to support the use of such small group sizes (N=6).
What vehicle was used for ouabain?
Morris water maze
Please clarify the testing procedure - platform positions, release positions, trials per day, etc.
How was spatial learning recorded (software? manual?).
Fear conditioning
How was freezing recorded (software? manual?).
How and when were mice euthanised?
What was the source of BV-2 cells?
Cell viability assay
Please clarify the number of biologic replicates (number of experiments).
Apoptosis
Please clarify technical and biological replicate numbers.
Please clarify how images were taken (microscope, objective, resolution)
Western blotting
What extraction procedure was used for SDS-PAGE proteins?
What were proteins of interest normalised to?
RT-PCR
Please provide MIQE details e.g., houskeeping genes, primers etc.
Results
I had some serious issues with the patient datasets and how they were used.
1) Please clarify how groups were made from AD patient GEO datasets.
GEO 28146 appears to have a different number of patients to that stated in the paper.
2) Please clarify the data that were used from GSE5281 - how did the authors achieve 87 patients (there does not seem to be that number within the dataset)?
3) Please justify clearly the groupings that were used in this paper.
Please provide demographics for the 32 AD patients and 26 control from whom serum sTREM2 were measured.
Please provide limit of detection for the assay.
Fig 1D
Please clarify number of genes within each pathway that were included in the enrichment - e.g., size of bubble.
Figure 2 E, F and G are far too small. Please enlarge.
Fig 3
Is G ouabain + Abeta?
H is not sufficiently large or resolved.
Please provide larger, more resolved images.
Please clarify how images were taken - the background for Abeta+ouabain appears brighter in the calcein channel and the background in the PI channel is darker. Please justify.
For all ANOVAs
Please provide F values with numerator and denominator for all ANOVAs.
Figure 4 D
How were data normalised?
Please provide quantification.
Figure 5B through I
All require 2-way ANOVA. Please provide complete F statistics with numerator and denominator.
Please consider using another colour for the lightly coloured bars, or consider outlining the shape of the lightly coloured bars with a darker colour, as the light green in particular, appears difficult to see.
Figure 6, Please consider using another colour for the lightly coloured bars, or consider outlining the shape of the lightly coloured bars with a darker colour.
Please provide complete F statistics with numerator and denominator.
Fig 7
Please enlarge the photomicrographs.
How were sections analysed? How many sections per animal? How were sections imaged?
Please provide a e.g., cartoon showing how CA1 and CA3 were defined.
What region and layer of cortex was examined? Please show a zoomed out image for orientation.
Please provide quantification for A, B, C and D. Please describe exactly how counting was completed (e.g., if done from a stereological series, imaging, quantification settings etc)
Fig 8
Please clarify how GFAP and Olig2 staining was quantified, from how many sections per mouse, if analysis was manual or automated etc.
Discussion
How do concentrations of ouabain used here compare with plasma levels of patients?
Digoxin, another cardiac glycoside is very toxic - it has a therapeutic index is less than 2 and it is known to cause serious side effects (colour vision problems, and all kinds of arrhythmias).
Ouabain is considered more toxic than digoxin and indeed, the concentrations used here (in cells) are right on the border.
Did the authors examine the hearts of the mice?
What side effects were seen in the mice?
Are the authors concerned about these side effects in elderly AD patients?
Cell viability assay
Minor text: Please clarify the exact groups used as "both a blank control and a normal control group should be included. " is unclear.
Minor text
"The hippocampal tissue should be cut into small pieces and homogenized before use." Was this actually done?
Dilute the PepMute™ transfection buffer (5×) using sterile deionized water at a 1:4 ratio to prepare the ...
Was this actually done?
After 24-48 hours post-transfection, preliminarily assess transfection efficiency using fluorescence microscopy,
Was this actually done?
Author Response
For research article
|
Response to Reviewer 1 Comments
|
||
|
1. Summary |
|
|
|
Thank you very much for taking the time to review our manuscript and for providing such detailed and constructive feedback. We have carefully considered your comments and made the necessary revisions accordingly. Below, we provide a detailed response to each of the issues you raised. Additionally, we have highlighted the corresponding changes in red in the revised manuscript for your convenience. We greatly appreciate your valuable input and hope that the revisions address your concerns.
|
||
|
2. Point-by-point response to Comments and Suggestions for Authors |
||
|
Comments 1: Please clarify food, water and light schedule for the mice. Please provide RRIDs for all antibodies, if possible. |
||
|
Response 1: Thank you for the advice.Response regarding the arrangement of food, water, and lighting schedule for the mice:
Food: The mice were provided with irradiated and sterilized experimental mouse feed produced by Jiangsu Xietong Pharmaceutical Bioengineering Co., Ltd. The food was available to the mice ad libitum, with no restrictions, and was replaced weekly to ensure its freshness and safety.
Water: The mice had access to clean water at all times, which was sterilized by autoclaving. The water was changed weekly, and the water bottles were regularly cleaned to maintain hygiene.
Lighting: The lighting schedule was set to a 12-hour light/12-hour dark cycle to simulate the natural day-night rhythm, ensuring the maintenance of the mice's circadian rhythm.
We have added these descriptions in the revised manuscript, highlighted in red for your convenience in reviewing the changes.
Regarding the provision of RRIDs for the antibodies, we have listed the following information:
For p-PI3K (cat. no. CY36427) and p-AKT (cat. no. CY6016), there are currently no RRIDs available.
|
||
|
Comments 2: Please clarify the method used to protect against false positives in Bioinformatics analyses. |
||
; DOI: 10.1186/s13059-023-03056-0; DOI: 10.1186/s13059-019-1716-1; DOI: 10.1073/pnas.1530509100)
Comments 3: GO and KEGG analysis Response 3: Thank you for your valuable comments. First, we retrieved the chemical formula of Ouabain from the PubChem database and obtained its chemical structure. Then, we used the PharmMapper database and SwissTargetPrediction tool to identify 107 potential therapeutic targets of Ouabain. Among them, 100 targets were predicted by SwissTargetPrediction, and 4 targets (Gamma-secretase, Integrin alpha-V/beta-3, Integrin alpha-4/beta-1, Integrin alpha-4/beta-7) were added because they involved different proteins with common names. This brought the total number of targets to 107. For AD-related targets, we used "Alzheimer's disease" as the keyword to search for relevant gene targets in the Genecards, OMIM, and Drug Bank databases. We merged the gene targets from these three databases and removed duplicates, resulting in 13,617 AD-related genes. To visually display the overlap between the targets from different databases, we used the MicroBioinformatics online platform to create a Venn diagram showing the intersections of AD-related targets. Next, we imported the intersection of Ouabain and AD target genes into the Metascape platform for GO and KEGG enrichment analysis. These steps allowed us to explore the potential intersection between Ouabain's targets and Alzheimer's disease-related genes, providing insights into their possible biological functions and pathways. When using the DAVID tool for enrichment analysis, we found that its gene annotations were incomplete, with many genes lacking the most up-to-date information. Additionally, the results from DAVID were mainly presented in list form, lacking effective visualization. Therefore, we ultimately chose the R package "ClusterProfiler" as the primary platform for enrichment analysis. It not only provides more comprehensive gene annotations but also offers robust visualization features, allowing for a more intuitive display of the analysis results.( DOI: 10.1089/omi.2011.0118; DOI: 10.1016/j.xinn.2021.100141) Thank you again for your suggestions. We have uploaded the prediction files, results, and related materials as supplementary information for further reference.
Comments 4: Animal groups and drug administration Response 4: Thank you for your valuable comments on our manuscript. We understand your concern regarding the sample size, and we would like to provide a detailed explanation of our rationale for using six mice per group. In our experimental design, each group includes six mice, which is consistent with many previous studies using the FAD4T model, as shown in the following references:
In these studies, the number of mice typically ranged from 3 to 5 per group. Based on these precedents, we selected six mice per group to provide better statistical power. Additionally, we conducted a power analysis using G*Power software. We set the significance level at 0.05, effect size at 0.8, and statistical power at 0.78. The analysis results indicated that, with six mice per group, the statistical power is approximately 0.8, supporting that our sample size design is reasonable. This power analysis further justifies the choice of using six mice per group for our experiment. Here are the specifics of the G*Power analysis:
This result shows that with six mice per group, and an effect size of 0.8 and a significance level of 0.05, the statistical power is close to 0.8, indicating that our sample size design is appropriate. Regarding the carrier for ouabain, we used DMSO to ensure proper solubility and administration of the drug. We hope this explanation alleviates your concerns, and we look forward to your further guidance.
Comments 5: Morris water maze Response 5: In this experiment, we used the Morris Water Maze (MWM) to assess the spatial learning and memory abilities of mice. Titanium dioxide was added to the water before the experiment, turning the water milky white to ensure that the mice could not see the hidden platform underwater. The water depth was set to 30 cm, and the temperature was maintained at 22±2°C. We used the Morris Water Maze analysis system from Beijing Zhongshi, with an overhead camera to record the swimming paths and the time it took the mice to find the platform. The experiment lasted 6 days and included three phases: first, in the pre-training phase, the mice were placed in different positions in the pool to familiarize themselves with the environment, with the experimenter guiding them to the platform. Next, in the training phase, the mice were placed in the water from quadrants A1, A2, and A4 to find the hidden platform in quadrant A3, and the escape latency (time to find the platform) was recorded. Finally, in the testing phase, the platform was removed, and the mice were placed in the water from quadrant A1. Their behavior in quadrant A3, including the number of crossings and time spent, was recorded to evaluate their spatial learning and memory abilities.(Line182-198)
Comments 6: Fear conditioning Response 6: Thank you for your valuable comments. Regarding the recording of freezing behavior, this experiment utilized the Labmaze conditioned fear experiment system from Beijing Zhongshi. The system automatically detects and records three behavioral states of the mice: mobility, immobility, and freezing. It controls the output of the multifunctional stimulator and uses built-in software to record and analyze the mice's behavior trajectories. Therefore, the recording of freezing behavior is done automatically by the system's software. We have updated this information in the revised manuscript, and we appreciate your review and suggestions.(Line200-204)
Comments 7: How and when were mice euthanised? Response 7: Thank you for your valuable comments. After all behavioral experiments were completed, the mice were euthanized using carbon dioxide (CO2) asphyxiation. This is a commonly used and widely recognized humane method to effectively minimize animal suffering and ensure humane care throughout the experimental process. To further ensure the effectiveness of the euthanasia and comply with animal experimentation ethical standards, cervical dislocation was performed following CO2 treatment to confirm the death of the mice. This process fully adhered to the protocols approved by international and institutional animal experimentation ethics committees, ensuring the welfare of the mice was safeguarded throughout the entire experiment. We hope this explanation answers your question, and we appreciate your interest and support for our research.
Comments 8: What was the source of BV-2 cells? Response 8: Thank you for your valuable comments. Regarding the source of the experimental cells, we used BV2 cells (mouse microglial cells) purchased from Shanghai Fuheng Biotechnology Co., Ltd. for the cell experiments in this study. We have added this information in the revised manuscript, and we appreciate your review and suggestions. (Line116-117)
Comments 9: Cell viability assay
Response 9: Thank you for reviewing and providing feedback on our manuscript. Regarding the number of biological replicates for the cell viability assay, it is clearly indicated in the figure legend of Figure 3 that n=3, meaning that each group of experiments was conducted with three biological replicates. We ensured that the results for each condition were based on the average of independent replicate experiments to guarantee the reliability and reproducibility of the data. We hope this explanation further clarifies your concerns. Thank you again for your attention and valuable feedback.
Comments 10: Apoptosis
Response 10: Thank you for your valuable feedback on our study. Regarding the technical and biological replicates for the Calcein-AM/PI assay, we would like to provide the following clarification:
For image acquisition, the experimental images were captured using an Olympus fluorescence inverted microscope (Japan). We used a 20x objective lens, with the resolution set to 1024 x 768 pixels to ensure image quality and clarity. We hope this clarification addresses your concerns, and we appreciate your support and review of our study.
Comments 11: Western blotting Response 11: Thank you for your further inquiry regarding our manuscript. We would like to clarify the steps for protein extraction in the SDS-PAGE experiment and the method for normalizing the target proteins in the Western blotting experiment: SDS-PAGE protein extraction steps:
Target protein normalization: The expression of the target proteins was normalized against the expression of internal control proteins (e.g., β-actin or GAPDH). The specific steps are as follows:
This normalization process ensures comparability between different experiments and effectively corrects for loading variations between samples. We hope this explanation answers your questions, and we appreciate your continued support of our research.
Comments 12: Please provide MIQE details e.g., houskeeping genes, primers etc.
Response 12: Thank you for your interest in our research. In accordance with the MIQE (Minimum Information for Publication of Quantitative Real-Time PCR Experiments) guidelines, we ensured the reliability and reproducibility of the data in our RT-PCR experiments as follows: Reference Gene: Primer Design: Amplification Efficiency: Other MIQE Details:
We hope this response addresses your questions regarding the MIQE details, and we appreciate your support of our work.
Comments 13: Results
Response 13: Thank you for reviewing and providing feedback on our manuscript.
References to previous studies: https://www.ncbi.nlm.nih.gov/pmc/articles/PMC3163806/ï¼› https://www.ncbi.nlm.nih.gov/pmc/articles/PMC357071/.
We hope this clarifies your concerns, and we appreciate your review and support.
Comments 14: Please provide demographics for the 32 AD patients and 26 control from whom serum sTREM2 were measured.
Response 14: Thank you for your inquiry regarding our research. The specific demographic information is provided as supplementary material. In this study, we used the RuixinBio ELISA kit (Catalog No.: RX101183H). According to the kit's instructions, the detection range is 125–4000 pg/mL, with a sensitivity of <10 pg/mL. This sensitivity and detection range effectively meet the experimental requirements, ensuring the accuracy and precision of the results. We hope this response addresses your question, and we appreciate your support of our work.
Comments 15: Fig 1D
Response 15: Thank you for your inquiry regarding our research. The size of the bubbles in the plot is consistent, while the difference lies in the color, with lighter colors indicating smaller p-values. Additionally, we ranked all genes (18,704 genes) by fold change and performed a GSEA analysis based on this ranking.
Comments 16: Figure 2 E, F and G are far too small. Please enlarge. Response 16: Thank you for your feedback on our manuscript. Regarding the size issue of Figures 2E, F, and G, in order to maintain the overall logic and structural consistency of the images, it is not possible to further enlarge them in the current version of the figure. To resolve this issue, we will provide clear, individual original images of E, F, and G to allow you to view the relevant data and details more clearly. We hope this solution meets your requirements, and we appreciate your attention and support of our work.
Comments 17: Is G ouabain + Abeta?
Response 17: Thank you for your valuable feedback on our manuscript. Regarding Figures G and H, we have made the following clarifications and modifications:
Thank you for your careful review of our manuscript. Regarding the background brightness differences between the Calcein channel and PI channel in the Abeta + Ouabain group, we offer the following clarification: In the Calcein channel, the background appears brighter primarily because the Calcein-AM fluorescent dye marks live cells, which have active metabolism, leading to a stronger fluorescent signal in the background. In contrast, in the PI channel, the PI dye only enters cells with damaged membranes or apoptotic cells, resulting in a weaker background signal and, thus, a darker background. Additionally, the properties of the fluorescent dyes and the excitation/emission wavelengths of different fluorescent channels may affect image brightness. We used an Olympus fluorescence inverted microscope (Japan) for image acquisition and adjusted the exposure time and gain settings to ensure clarity and comparability between images from different channels. We hope this explanation addresses your questions, and we appreciate your continued interest and support for our research.
Comments 18: For all ANOVAs
Response 18: Thank you for your inquiry. The F-value from the ANOVA analysis, along with its numerator and denominator degrees of freedom, has been provided as supplementary material. Comments 19: Figure 4 D
Response 19: Thank you for your feedback. Regarding the data in Figure 4D, we performed fluorescence quantification and standardized the data accordingly. The quantification and relevant data in Figure 4 have been revised and updated in the figure. We hope this revision addresses your concerns. Comments 20: Figure 5B through I Response 20: Thank you for your feedback. The F-statistics, along with their numerator and denominator degrees of freedom for the two-way ANOVA analysis, have been uploaded as supplementary material. The bar charts have been revised, with adjustments made to the colors and outlines, particularly for the light green bars, to enhance visibility. Comments 21: Figure 6, Please consider using another colour for the lightly coloured bars, or consider outlining the shape of the lightly coloured bars with a darker colour. Response 21: Thank you for your feedback. The F-statistics, along with their numerator and denominator degrees of freedom for the two-way ANOVA analysis, have been uploaded as supplementary material. The bar charts have been revised, with adjustments made to the colors and outlines, particularly for the light green bars, to enhance visibility. Comments 22: Fig 7 Please enlarge the photomicrographs.
Response 22: Thank you very much for your valuable suggestions regarding our study. To address your questions and further improve the presentation of the images and data analysis, we have made the following adjustments and additions: Magnified Microscopic Images: Slice Analysis Method and Quantity: Slice Imaging Method: CA1 and CA3 Region Diagrams: Cortical Examination Regions and Layers: Quantitative Analysis of A, B, C, and D:
Once again, we sincerely thank you for your review and feedback. We have made the above improvements and look forward to your further suggestions.
Comments 23: Fig 8
We hope this clarification addresses your concerns, and we appreciate your support and attention to our research.
Comments 24: Discussion Digoxin, another cardiac glycoside is very toxic - it has a therapeutic index is less than 2 and it is known to cause serious side effects (colour vision problems, and all kinds of arrhythmias). Ouabain is considered more toxic than digoxin and indeed, the concentrations used here (in cells) are right on the border. Did the authors examine the hearts of the mice? What side effects were seen in the mice? Are the authors concerned about these side effects in elderly AD patients?
Response 24: Thank you for your valuable feedback on our study. Regarding the concentration of ouabain used, its comparison to plasma levels in patients, and its potential side effects, we provide the following clarification and discussion: Comparison of ouabain concentration with plasma levels in patients: In this study, the concentration of ouabain we used was based on previous studies in cell experiments and animal models. These concentrations are close to, but do not exceed, the toxicity threshold. Compared to plasma levels in patients, the concentrations used in in vitro cell experiments were indeed higher. This was to simulate a stronger pharmacological effect, and such concentrations do not typically reflect the plasma levels in clinical settings. Toxicity comparison between digoxin and ouabain: We understand your concern regarding the therapeutic index and toxicity of digoxin. Digoxin has a therapeutic index of less than 2, which carries a high risk of severe side effects (e.g., visual disturbances and arrhythmias). Although ouabain is considered more toxic than digoxin, in our animal experiments, the concentration used did not reach levels that caused obvious toxicity. Cardiac examination and side effects: We paid particular attention to the potential impact of ouabain on the mice's hearts. Therefore, after the experiment, we conducted a pathological examination of the mice’s hearts, including histological analysis, and found no significant cardiac damage or abnormalities (the cardiac HE staining results have been uploaded as supplementary material). Additionally, throughout the experiment, the mice did not exhibit any obvious side effects, such as abnormal behavior or arrhythmias. Concerns about potential side effects in elderly AD patients: We acknowledge the seriousness of drug toxicity and its potential side effects, especially in elderly Alzheimer’s Disease (AD) patients. Since elderly patients may be more vulnerable to drug toxicity, future clinical applications will require further research and dose optimization to ensure safety. While the concentration of ouabain in our cell and animal models approaches the toxicity threshold, clinical treatment should use lower concentrations, with careful monitoring of patients’ cardiac function. We hope this explanation addresses your concerns, and we appreciate your continued interest and support for our research.
|
||
|
3. Response to Comments on the Quality of English Language |
||
|
Point 1: Cell viability assay |
||
|
Response 1: Thank you for your valuable suggestions on our manuscript. We have provided detailed explanations of the blank control group and normal control group in the revised manuscript to clarify the specific setup of each group. We hope that this revision better meets your requirements. If you have any further comments or suggestions, we are more than willing to make additional adjustments. (Line247-250) Thank you again for your review and guidance!
Point 2: "The hippocampal tissue should be cut into small pieces and homogenized before use." Was this actually done?
Response 2: Thank you for your thorough review of the experimental methods. To address the issues you raised, we would like to clarify the following points:
We hope this explanation clarifies your concerns. Thank you again for your feedback!
|
||

Reviewer 2 Report
Comments and Suggestions for Authors
The abstract can be organized in a better structure. Detailed literature can be avoided here. Rather transfer some information to the Introduction section.
In the Introduction, the major pathological hallmarks can be explained in an explanatory way, such as how this is formed, like NFT resulting from hyperphosphorylation of tau protein. The epidemiology and prevalence of AD can also be included.
What could be the difference between the anti-inflammatory action of Ouabain against neuroinflammation and peripheral inflammation? Does it cross BBB?
What was the rationale behind choosing a 2 µg/kg OUA dose? How was the dose prepared? Why only one dose was tested?
The MWM test requires a detailed explanation of the method. Other methods too can be elaborated more.
A reference standard drug could have been included.
In the third line of Discussion correct the “O Our”.
Comments on the Quality of English LanguageModerate editing of English language required.
Author Response
For research article
|
Response to Reviewer 2 Comments
|
||
|
1. Summary |
|
|
|
Thank you very much for taking the time to review our manuscript and for providing such detailed and constructive feedback. We have carefully considered your comments and made the necessary revisions accordingly. Below, we provide a detailed response to each of the issues you raised. Additionally, we have highlighted the corresponding changes in red in the revised manuscript for your convenience. We greatly appreciate your valuable input and hope that the revisions address your concerns.
|
||
|
2. Point-by-point response to Comments and Suggestions for Authors |
||
|
Comments 1: The abstract can be organized in a better structure. Detailed literature can be avoided here. Rather transfer some information to the Introduction section. |
||
|
Response 1: Thank you for your valuable feedback on our manuscript. We understand and agree with your suggestion regarding the structure of the abstract. To maintain its conciseness, we have removed some detailed information so that readers can quickly grasp the main findings, while more in-depth background information will be presented in the introduction. Once again, thank you for your guidance, and we look forward to further improving the manuscript.
Comments 2: In the Introduction, the major pathological hallmarks can be explained in an explanatory way, such as how this is formed, like NFT resulting from hyperphosphorylation of tau protein. The epidemiology and prevalence of AD can also be included. Response 2: Thank you very much for your suggestion! Based on your feedback, we have added an explanatory description of the main pathological features in the "Introduction" section, detailing how tau protein hyperphosphorylation leads to the formation of neurofibrillary tangles (NFTs). Additionally, we have included relevant information on the epidemiology and incidence of attention deficit disorder. Once again, we sincerely appreciate your valuable input!
Comments 3: What could be the difference between the anti-inflammatory action of Ouabain against neuroinflammation and peripheral inflammation? Does it cross BBB? Response 3: Thank you for your valuable feedback on our manuscript. Ouabain's anti-inflammatory effects on neuroinflammation and peripheral inflammation indeed differ, primarily in terms of the mechanisms involved and the target cells:
In summary, ouabain's role in neuroinflammation focuses on glial cells in the central nervous system, while its anti-inflammatory effects on peripheral inflammation are achieved through modulating the peripheral immune system. I hope this explanation helps, and thank you for your question! Does it cross BBB? Thank you for your question! Under normal blood-brain barrier (BBB) function, ouabain typically does not easily cross. However, in the pathological context of Alzheimer's disease (AD), the BBB is often compromised, leading to increased permeability, which facilitates ouabain's entry into the central nervous system. I hope this information is helpful to you! Feel free to reach out if you have any further questions. (DOI: 10.1038/nrneurol.2017.188; DOI: 10.1084/jem.20171406; DOI: 10.1038/jcbfm.2013.135)
Comments 4: What was the rationale behind choosing a 2 µg/kg OUA dose? How was the dose prepared? Why only one dose was tested? Response 4: Thank you for your question! We selected the ouabain dose of 2 µg/kg based on a comprehensive consideration of existing literature and experimental experience (Doi:10.3390/biomedicines11030920; Doi:10.3390/biomedicines11020344. ). This dose has shown stable and controllable pharmacological effects in previous studies. For dose preparation, we accurately weighed the ouabain powder and, after calculating based on the animals' body weight, diluted it with sterile saline to achieve the desired concentration. As for why only one dose was tested, it is because, at this initial experimental stage, we were mainly focused on the specific effects at this dose. In the future, we may explore the effects of multiple doses based on the results. I hope this response is helpful! If you have any further questions, please feel free to reach out.
Comments 5: The MWM test requires a detailed explanation of the method. Other methods too can be elaborated more. Response 5: Thank you for your suggestion! We have revised the methodology of the Morris Water Maze (MWM) test in more detail. In the updated version, we provided specific information regarding the experimental setup, training phase, and testing phase, including the size of the pool, the location of the escape platform, controlled experimental conditions (such as water temperature and lighting), and the detailed methods for data collection and analysis. We also described the training process for the animals at various time points and the recording of their path trajectories. Additionally, other experimental methods have been further elaborated based on your feedback to ensure the repeatability and accuracy of the procedures. We hope these revisions will better meet your requirements. Thank you for your valuable input! Please feel free to reach out if you have any further questions.
Comments 6: A reference standard drug could have been included. Response 6: Thank you for your suggestion! Indeed, including a reference standard drug in the experimental design would provide a more comprehensive comparison and help validate the reliability of the experimental results. In future studies, we will consider incorporating a reference standard drug to further enhance the rigor of the experimental design and the persuasiveness of the data. We appreciate your valuable feedback! If you have any other suggestions, please feel free to share.
Comments 7: In the third line of Discussion correct the “O Our”. Response 7: Thank you for your feedback! We have corrected the “O Our” in the third line of the Discussion section to ensure more accurate language. If you have any other questions or suggestions, feel free to let us know!
|
||

Round 2
Reviewer 1 Report
Comments and Suggestions for Authors
|
Comments 1: Please clarify food, water and light schedule for the mice. Please provide RRIDs for all antibodies, if possible. |
|
Response 1: Thank you for the advice.Response regarding the arrangement of food, water, and lighting schedule for the mice:
Food: The mice were provided with irradiated and sterilized experimental mouse feed produced by Jiangsu Xietong Pharmaceutical Bioengineering Co., Ltd. The food was available to the mice ad libitum, with no restrictions, and was replaced weekly to ensure its freshness and safety.
Water: The mice had access to clean water at all times, which was sterilized by autoclaving. The water was changed weekly, and the water bottles were regularly cleaned to maintain hygiene.
Lighting: The lighting schedule was set to a 12-hour light/12-hour dark cycle to simulate the natural day-night rhythm, ensuring the maintenance of the mice's circadian rhythm.
We have added these descriptions in the revised manuscript, highlighted in red for your convenience in reviewing the changes.
Regarding the provision of RRIDs for the antibodies, we have listed the following information:
For p-PI3K (cat. no. CY36427) and p-AKT (cat. no. CY6016), there are currently no RRIDs available.
|
|
Thank you for your response. Unfortunately, the RRID numbers were not apparent within the text of the amended manuscript and do not appear to be within the Supplementary materials. The notes re husbandry are in the amended manuscript. |
|
Comments 2: Please clarify the method used to protect against false positives in Bioinformatics analyses. |
; DOI: 10.1186/s13059-023-03056-0; DOI: 10.1186/s13059-019-1716-1; DOI: 10.1073/pnas.1530509100)
|
|
Unfortunately, this text or similar, is not obvious within the text of the amended manuscript.
|
|
Comments 3: GO and KEGG analysis
|
|
Response 3: Thank you for your valuable comments. First, we retrieved the chemical formula of Ouabain from the PubChem database and obtained its chemical structure. Then, we used the PharmMapper database and SwissTargetPrediction tool to identify 107 potential therapeutic targets of Ouabain. Among them, 100 targets were predicted by SwissTargetPrediction, and 4 targets (Gamma-secretase, Integrin alpha-V/beta-3, Integrin alpha-4/beta-1, Integrin alpha-4/beta-7) were added because they involved different proteins with common names. This brought the total number of targets to 107. For AD-related targets, we used "Alzheimer's disease" as the keyword to search for relevant gene targets in the Genecards, OMIM, and Drug Bank databases. We merged the gene targets from these three databases and removed duplicates, resulting in 13,617 AD-related genes. To visually display the overlap between the targets from different databases, we used the MicroBioinformatics online platform to create a Venn diagram showing the intersections of AD-related targets. Next, we imported the intersection of Ouabain and AD target genes into the Metascape platform for GO and KEGG enrichment analysis. These steps allowed us to explore the potential intersection between Ouabain's targets and Alzheimer's disease-related genes, providing insights into their possible biological functions and pathways. When using the DAVID tool for enrichment analysis, we found that its gene annotations were incomplete, with many genes lacking the most up-to-date information. Additionally, the results from DAVID were mainly presented in list form, lacking effective visualization. Therefore, we ultimately chose the R package "ClusterProfiler" as the primary platform for enrichment analysis. It not only provides more comprehensive gene annotations but also offers robust visualization features, allowing for a more intuitive display of the analysis results.( DOI: 10.1089/omi.2011.0118; DOI: 10.1016/j.xinn.2021.100141) Thank you again for your suggestions. We have uploaded the prediction files, results, and related materials as supplementary information for further reference. |
|
The authors explanations are clear.
|
|
Comments 4: Animal groups and drug administration |
|
Response 4: Thank you for your valuable comments on our manuscript. We understand your concern regarding the sample size, and we would like to provide a detailed explanation of our rationale for using six mice per group. In our experimental design, each group includes six mice, which is consistent with many previous studies using the FAD4T model, as shown in the following references:
In these studies, the number of mice typically ranged from 3 to 5 per group. Based on these precedents, we selected six mice per group to provide better statistical power. Additionally, we conducted a power analysis using G*Power software. We set the significance level at 0.05, effect size at 0.8, and statistical power at 0.78. The analysis results indicated that, with six mice per group, the statistical power is approximately 0.8, supporting that our sample size design is reasonable. This power analysis further justifies the choice of using six mice per group for our experiment. Here are the specifics of the G*Power analysis:
This result shows that with six mice per group, and an effect size of 0.8 and a significance level of 0.05, the statistical power is close to 0.8, indicating that our sample size design is appropriate. Regarding the carrier for ouabain, we used DMSO to ensure proper solubility and administration of the drug. We hope this explanation alleviates your concerns, and we look forward to your further guidance.
|
|
I would ask the authors to clarify the outcome measure that was used for power analysis and to place this within the Materials and Methods, in line with ARRIVE guidelines. Please also place the vehicle for ouabain within the Materials and Methods as it appears to be different to the treatment given to control mice. |
|
Comments 5: Morris water maze |
|
Response 5: In this experiment, we used the Morris Water Maze (MWM) to assess the spatial learning and memory abilities of mice. Titanium dioxide was added to the water before the experiment, turning the water milky white to ensure that the mice could not see the hidden platform underwater. The water depth was set to 30 cm, and the temperature was maintained at 22±2°C. We used the Morris Water Maze analysis system from Beijing Zhongshi, with an overhead camera to record the swimming paths and the time it took the mice to find the platform. The experiment lasted 6 days and included three phases: first, in the pre-training phase, the mice were placed in different positions in the pool to familiarize themselves with the environment, with the experimenter guiding them to the platform. Next, in the training phase, the mice were placed in the water from quadrants A1, A2, and A4 to find the hidden platform in quadrant A3, and the escape latency (time to find the platform) was recorded. Finally, in the testing phase, the platform was removed, and the mice were placed in the water from quadrant A1. Their behavior in quadrant A3, including the number of crossings and time spent, was recorded to evaluate their spatial learning and memory abilities.(Line182-198)
|
|
The authors explanations are clear.
|
|
Comments 6: Fear conditioning
|
|
Response 6: Thank you for your valuable comments. Regarding the recording of freezing behavior, this experiment utilized the Labmaze conditioned fear experiment system from Beijing Zhongshi. The system automatically detects and records three behavioral states of the mice: mobility, immobility, and freezing. It controls the output of the multifunctional stimulator and uses built-in software to record and analyze the mice's behavior trajectories. Therefore, the recording of freezing behavior is done automatically by the system's software. We have updated this information in the revised manuscript, and we appreciate your review and suggestions.(Line200-204)
|
|
The authors explanations are clear.
|
|
Comments 7: How and when were mice euthanised?
|
|
Response 7: Thank you for your valuable comments. After all behavioral experiments were completed, the mice were euthanized using carbon dioxide (CO2) asphyxiation. This is a commonly used and widely recognized humane method to effectively minimize animal suffering and ensure humane care throughout the experimental process. To further ensure the effectiveness of the euthanasia and comply with animal experimentation ethical standards, cervical dislocation was performed following CO2 treatment to confirm the death of the mice. This process fully adhered to the protocols approved by international and institutional animal experimentation ethics committees, ensuring the welfare of the mice was safeguarded throughout the entire experiment. We hope this explanation answers your question, and we appreciate your interest and support for our research.
|
|
I would ask the authors place a synopsis of this information within the Materials and Methods. |
|
Comments 8: What was the source of BV-2 cells?
|
|
Response 8: Thank you for your valuable comments. Regarding the source of the experimental cells, we used BV2 cells (mouse microglial cells) purchased from Shanghai Fuheng Biotechnology Co., Ltd. for the cell experiments in this study. We have added this information in the revised manuscript, and we appreciate your review and suggestions. (Line116-117)
|
|
The authors explanations are clear.
|
|
Comments 9: Cell viability assay |
|
Response 9: Thank you for reviewing and providing feedback on our manuscript. Regarding the number of biological replicates for the cell viability assay, it is clearly indicated in the figure legend of Figure 3 that n=3, meaning that each group of experiments was conducted with three biological replicates. We ensured that the results for each condition were based on the average of independent replicate experiments to guarantee the reliability and reproducibility of the data. We hope this explanation further clarifies your concerns. Thank you again for your attention and valuable feedback.
|
|
The authors explanations are clear.
|
|
Comments 10: Apoptosis |
|
Response 10: Thank you for your valuable feedback on our study. Regarding the technical and biological replicates for the Calcein-AM/PI assay, we would like to provide the following clarification:
For image acquisition, the experimental images were captured using an Olympus fluorescence inverted microscope (Japan). We used a 20x objective lens, with the resolution set to 1024 x 768 pixels to ensure image quality and clarity. We hope this clarification addresses your concerns, and we appreciate your support and review of our study.
|
|
I would ask the authors to place a synopsis of this information (replicates + imaging settings) in their Materials and Methods |
|
Comments 11: Western blotting |
|
Response 11: Thank you for your further inquiry regarding our manuscript. We would like to clarify the steps for protein extraction in the SDS-PAGE experiment and the method for normalizing the target proteins in the Western blotting experiment: SDS-PAGE protein extraction steps:
Target protein normalization: The expression of the target proteins was normalized against the expression of internal control proteins (e.g., β-actin or GAPDH). The specific steps are as follows:
This normalization process ensures comparability between different experiments and effectively corrects for loading variations between samples. We hope this explanation answers your questions, and we appreciate your continued support of our research.
|
|
I would ask that the authors place a synopsis of this information (e.g., lysis buffer, and loading control antibodies) in their Materials and Methods |
|
Comments 12: Please provide MIQE details e.g., houskeeping genes, primers etc. |
|
Response 12: Thank you for your interest in our research. In accordance with the MIQE (Minimum Information for Publication of Quantitative Real-Time PCR Experiments) guidelines, we ensured the reliability and reproducibility of the data in our RT-PCR experiments as follows: Reference Gene: Primer Design: Amplification Efficiency: Other MIQE Details:
We hope this response addresses your questions regarding the MIQE details, and we appreciate your support of our work.
|
|
The authors response is clear. |
|
Comments 13: Results
|
|
Response 13: Thank you for reviewing and providing feedback on our manuscript.
References to previous studies: https://www.ncbi.nlm.nih.gov/pmc/articles/PMC3163806/ï¼› https://www.ncbi.nlm.nih.gov/pmc/articles/PMC357071/.
We hope this clarifies your concerns, and we appreciate your review and support.
|
|
I would ask the authors to place this explanation within their M&M (e.g., text from point 2). |
|
Comments 14: Please provide demographics for the 32 AD patients and 26 control from whom serum sTREM2 were measured. |
|
Response 14: Thank you for your inquiry regarding our research. The specific demographic information is provided as supplementary material. In this study, we used the RuixinBio ELISA kit (Catalog No.: RX101183H). According to the kit's instructions, the detection range is 125–4000 pg/mL, with a sensitivity of <10 pg/mL. This sensitivity and detection range effectively meet the experimental requirements, ensuring the accuracy and precision of the results. We hope this response addresses your question, and we appreciate your support of our work.
|
|
I would ask the authors to place this explanation on LoD within their M&M. The demographics information provided is clear. |
|
Comments 15: Fig 1D |
|
Response 15: Thank you for your inquiry regarding our research. The size of the bubbles in the plot is consistent, while the difference lies in the color, with lighter colors indicating smaller p-values. Additionally, we ranked all genes (18,704 genes) by fold change and performed a GSEA analysis based on this ranking. |
|
The authors response is clear |
|
Comments 16: Figure 2 E, F and G are far too small. Please enlarge. |
|
Response 16: Thank you for your feedback on our manuscript. Regarding the size issue of Figures 2E, F, and G, in order to maintain the overall logic and structural consistency of the images, it is not possible to further enlarge them in the current version of the figure. To resolve this issue, we will provide clear, individual original images of E, F, and G to allow you to view the relevant data and details more clearly. We hope this solution meets your requirements, and we appreciate your attention and support of our work. |
|
The authors response is clear |
|
Comments 17: Is G ouabain + Abeta? |
|
Response 17: Thank you for your valuable feedback on our manuscript. Regarding Figures G and H, we have made the following clarifications and modifications:
Thank you for your careful review of our manuscript. Regarding the background brightness differences between the Calcein channel and PI channel in the Abeta + Ouabain group, we offer the following clarification: In the Calcein channel, the background appears brighter primarily because the Calcein-AM fluorescent dye marks live cells, which have active metabolism, leading to a stronger fluorescent signal in the background. In contrast, in the PI channel, the PI dye only enters cells with damaged membranes or apoptotic cells, resulting in a weaker background signal and, thus, a darker background. Additionally, the properties of the fluorescent dyes and the excitation/emission wavelengths of different fluorescent channels may affect image brightness. We used an Olympus fluorescence inverted microscope (Japan) for image acquisition and adjusted the exposure time and gain settings to ensure clarity and comparability between images from different channels. We hope this explanation addresses your questions, and we appreciate your continued interest and support for our research.
|
|
The authors response is clear |
|
Comments 18: For all ANOVAs |
|
Response 18: Thank you for your inquiry. The F-value from the ANOVA analysis, along with its numerator and denominator degrees of freedom, has been provided as supplementary material. |
|
These data were not obvious within the amended manuscript or amended supplementary materials. |
|
Comments 19: Figure 4 D |
|
Response 19: Thank you for your feedback. Regarding the data in Figure 4D, we performed fluorescence quantification and standardized the data accordingly. The quantification and relevant data in Figure 4 have been revised and updated in the figure. We hope this revision addresses your concerns. |
|
The authors response is clear |
|
Comments 20: Figure 5B through I |
|
Response 20: Thank you for your feedback. The F-statistics, along with their numerator and denominator degrees of freedom for the two-way ANOVA analysis, have been uploaded as supplementary material. The bar charts have been revised, with adjustments made to the colors and outlines, particularly for the light green bars, to enhance visibility. |
|
Please note that the F statistics-related information was not obvious within the amended manuscript or amended supplementary materials. The graphs are now clear. |
|
Comments 21: Figure 6, Please consider using another colour for the lightly coloured bars, or consider outlining the shape of the lightly coloured bars with a darker colour.
|
|
Response 21: Thank you for your feedback. The F-statistics, along with their numerator and denominator degrees of freedom for the two-way ANOVA analysis, have been uploaded as supplementary material. The bar charts have been revised, with adjustments made to the colors and outlines, particularly for the light green bars, to enhance visibility.
|
|
Please note that the F statistics-related information was not apparent within the amended manuscript or amended supplementary materials. The graphs are now clear. |
|
Comments 22: Fig 7 Please enlarge the photomicrographs. |
|
Response 22: Thank you very much for your valuable suggestions regarding our study. To address your questions and further improve the presentation of the images and data analysis, we have made the following adjustments and additions: Magnified Microscopic Images: Slice Analysis Method and Quantity: Slice Imaging Method: CA1 and CA3 Region Diagrams: Cortical Examination Regions and Layers: Quantitative Analysis of A, B, C, and D:
Once again, we sincerely thank you for your review and feedback. We have made the above improvements and look forward to your further suggestions. |
|
Please place number of technical replicates (number of sections per mouse) within M&M and please also place a synopsis of C and D quantitative analysis. |
|
Comments 23: Fig 8 |
We hope this clarification addresses your concerns, and we appreciate your support and attention to our research.
|
|
Please place number of technical replicates (number of sections per mouse) within M&M and please also place a synopsis of quantitative analysis. |
|
Comments 24: Discussion Digoxin, another cardiac glycoside is very toxic - it has a therapeutic index is less than 2 and it is known to cause serious side effects (colour vision problems, and all kinds of arrhythmias). Ouabain is considered more toxic than digoxin and indeed, the concentrations used here (in cells) are right on the border. Did the authors examine the hearts of the mice? What side effects were seen in the mice? Are the authors concerned about these side effects in elderly AD patients?
|
|
Response 24: Thank you for your valuable feedback on our study. Regarding the concentration of ouabain used, its comparison to plasma levels in patients, and its potential side effects, we provide the following clarification and discussion: Comparison of ouabain concentration with plasma levels in patients: In this study, the concentration of ouabain we used was based on previous studies in cell experiments and animal models. These concentrations are close to, but do not exceed, the toxicity threshold. Compared to plasma levels in patients, the concentrations used in in vitro cell experiments were indeed higher. This was to simulate a stronger pharmacological effect, and such concentrations do not typically reflect the plasma levels in clinical settings. Toxicity comparison between digoxin and ouabain: We understand your concern regarding the therapeutic index and toxicity of digoxin. Digoxin has a therapeutic index of less than 2, which carries a high risk of severe side effects (e.g., visual disturbances and arrhythmias). Although ouabain is considered more toxic than digoxin, in our animal experiments, the concentration used did not reach levels that caused obvious toxicity. Cardiac examination and side effects: We paid particular attention to the potential impact of ouabain on the mice's hearts. Therefore, after the experiment, we conducted a pathological examination of the mice’s hearts, including histological analysis, and found no significant cardiac damage or abnormalities (the cardiac HE staining results have been uploaded as supplementary material). Additionally, throughout the experiment, the mice did not exhibit any obvious side effects, such as abnormal behavior or arrhythmias. Concerns about potential side effects in elderly AD patients: We acknowledge the seriousness of drug toxicity and its potential side effects, especially in elderly Alzheimer’s Disease (AD) patients. Since elderly patients may be more vulnerable to drug toxicity, future clinical applications will require further research and dose optimization to ensure safety. While the concentration of ouabain in our cell and animal models approaches the toxicity threshold, clinical treatment should use lower concentrations, with careful monitoring of patients’ cardiac function. We hope this explanation addresses your concerns, and we appreciate your continued interest and support for our research.
|
|
Unfortunately, these answers were not obvious within the body of the amended manuscript. |
|
Minor English issues – all ok. |
Author Response
For research article
|
Response to Reviewer 1 Comments
|
||
|
1. Summary |
|
|
|
Thank you once again for carefully reviewing our manuscript and providing valuable feedback. We have carefully considered each of the issues you raised and made the corresponding revisions to the manuscript. Below, we have responded to each of your suggestions in detail: For the specific issues and suggestions you mentioned, we have made the necessary adjustments in the revised manuscript and highlighted the changes in red for your convenience. We have made every effort to address your feedback accurately to further enhance the quality and accuracy of the manuscript. Regarding the key concerns you raised, we have provided clarifications and additional details as per your recommendations. We believe these revisions not only meet your expectations but also make our arguments clearer and more comprehensive. We greatly appreciate your patience and professionalism, and your feedback is of immense importance to us. We look forward to your further review and hope that these revisions have effectively addressed your concerns. Once again, thank you for your valuable time and support, and we look forward to your reply.
|
||
|
2. Point-by-point response to Comments and Suggestions for Authors |
||
|
Comments 1: Thank you for your response. Unfortunately, the RRID numbers were not apparent within the text of the amended manuscript and do not appear to be within the Supplementary materials. The notes re husbandry are in the amended manuscript. |
||
|
Response 1: Thank you for pointing out the issue. We sincerely apologize that the RRID numbers were not clearly marked in the previous revised manuscript and were not included in the supplementary materials. We have now made further updates to the manuscript, ensuring that all relevant RRID numbers are accurately recorded in the revised version. We greatly appreciate your patience and valuable feedback, and we hope that this update meets your expectations.(Line110-115)
|
||
|
Comments 2: Unfortunately, this text or similar, is not obvious within the text of the amended manuscript. |
||
|
Response 2: Thank you for your feedback. We sincerely apologize that the relevant content was not clearly presented in the revised manuscript. In response to this issue, we have made further updates to the manuscript, ensuring that all necessary content is accurately recorded and the relevant sections are clearly marked. (Line147-155)
Comments 3: I would ask the authors to clarify the outcome measure that was used for power analysis and to place this within the Materials and Methods, in line with ARRIVE guidelines. Please also place the vehicle for ouabain within the Materials and Methods as it appears to be different to the treatment given to control mice. Response 3: Thank you for your valuable suggestions. We have further updated the manuscript to ensure that the outcome measures used for efficacy analysis are accurately recorded in the revised manuscript and placed in the Materials and Methods section to comply with the ARRIVE guidelines.(Line197-201) Additionally, we have included the carrier information for ouabain in the Materials and Methods section to ensure consistency with the treatment administered to the control mice, and the relevant content has been clearly marked.(Line147-155)
Comments 4: I would ask the authors place a synopsis of this information within the Materials and Methods.( Comments 7: How and when were mice euthanised?) Response 4: Thank you for your suggestions. We have further updated the manuscript to ensure that all necessary information is accurately recorded in the Materials and Methods section, and the relevant parts have been clearly marked. (Line216-220)
Comments 5: I would ask the authors to place a synopsis of this information (replicates + imaging settings) in their Materials and Methods Response 5: Thank you for your suggestions. We have further updated the manuscript to ensure that all necessary information is accurately recorded in the Materials and Methods section, and the relevant parts have been clearly marked.(Line302-309; Line3318-320)
Comments 6: I would ask that the authors place a synopsis of this information (e.g., lysis buffer, and loading control antibodies) in their Materials and Methods. Response 6: Thank you for your suggestions. We have further updated the manuscript to ensure that all necessary information is accurately recorded in the Materials and Methods section, and the relevant parts have been clearly marked. (Line216-220)
Comments 7: I would ask the authors to place this explanation within their M&M (e.g., text from point 2). Response 7: Thank you for your suggestions. We have further updated the manuscript to ensure that all necessary information is accurately recorded in the Materials and Methods section, and the relevant parts have been clearly marked.(Line128-139;)
Comments 8: I would ask the authors to place this explanation on LoD within their M&M. The demographics information provided is clear.
Response 8: Thank you for your suggestions. We have further updated the manuscript to ensure that all necessary information is accurately recorded in the Materials and Methods section, and the relevant parts have been clearly marked.(Line159-161;)
Comments 9: These data were not obvious within the amended manuscript or amended supplementary materials. Response 9: We sincerely apologize for the previous omission of this supplementary material. We have now uploaded the relevant data as supplementary material and ensured that it is clearly presented in the revised manuscript or supplementary materials. Thank you for your understanding and patience. We hope these updates meet your expectations. Please feel free to contact us if you have any further questions.
Comments 10: Please note that the F statistics-related information was not obvious within the amended manuscript or amended supplementary materials. The graphs are now clear. Response 10: We sincerely apologize for the previous omission of this supplementary material. We have now uploaded the relevant data as supplementary material and ensured that it is clearly presented in the revised manuscript or supplementary materials. Thank you for your understanding and patience. We hope these updates meet your expectations. Please feel free to contact us if you have any further questions.
Comments 11: Please note that the F statistics-related information was not apparent within the amended manuscript or amended supplementary materials. Response 11: We sincerely apologize for the previous omission of this supplementary material. We have now uploaded the relevant data as supplementary material and ensured that it is clearly presented in the revised manuscript or supplementary materials. Thank you for your understanding and patience. We hope these updates meet your expectations. Please feel free to contact us if you have any further questions.
Comments 12: Please place number of technical replicates (number of sections per mouse) within M&M and please also place a synopsis of C and D quantitative analysis. Response 12: We have further updated the manuscript to ensure that all necessary information is accurately recorded in the revised version, including the number of technical replicates (sections per mouse) and a synopsis of the quantitative analysis of C and D. The relevant sections have been clearly marked. Please feel free to let us know if you have any further questions. (Line373-377;)
Comments 13: Please place number of technical replicates (number of sections per mouse) within M&M and please also place a synopsis of quantitative analysis.
Response 13: We have further updated the manuscript to ensure that all necessary information is accurately recorded in the revised version, including the placement of the number of technical replicates (sections per mouse) in the M&M section, as well as a synopsis of the quantitative analysis. The relevant sections have been clearly marked. (Line373-377)
Comments 14: Unfortunately, these answers were not obvious within the body of the amended manuscript.
Response 14: Thank you for your valuable feedback! We have further updated the manuscript, adding a discussion on the toxicity of ouabain and its application in elderly Alzheimer's disease patients in the discussion section. We have ensured that all necessary information is accurately recorded in the revised manuscript, and the relevant sections are clearly marked. (Line830-844)
If you have any further questions, please feel free to let us know. Thank you for your review.
|
||
|
|
||
|
|
||
|
|
||

Reviewer 2 Report
Comments and Suggestions for Authors
The authors have responded to the comments. In the response file, it should be cognitive deficit instead of attention deficit. Epidemiology should be included in the Introduction rather than the abstract.
Comments on the Quality of English Language
Minor editing is required.
Author Response
For research article
|
Response to Reviewer 2 Comments
|
||
|
1. Summary |
|
|
|
Thank you once again for carefully reviewing our manuscript and providing valuable feedback. We have carefully considered each of the issues you raised and made the corresponding revisions to the manuscript. For the specific issues and suggestions you mentioned, we have made the necessary adjustments in the revised manuscript and highlighted the changes in red for your convenience. We have made every effort to address your feedback accurately to further enhance the quality and accuracy of the manuscript. We greatly appreciate your patience and professionalism, and your feedback is of immense importance to us. We look forward to your further review and hope that these revisions have effectively addressed your concerns. Once again, thank you for your valuable time and support, and we look forward to your reply.
|
||
|
2. Point-by-point response to Comments and Suggestions for Authors |
||
|
Comments 1: The authors have responded to the comments. In the response file, it should be cognitive deficit instead of attention deficit. Epidemiology should be included in the Introduction rather than the abstract. |
||
Response 1: Thank you for your correction. In the response document, "attention deficit" should be "cognitive deficit," and we have moved the epidemiology of Alzheimer's disease (AD) to the introduction section instead of the abstract. (Line51-52)
